# Mellow: a small audio language model for reasoning

**Soham Deshmukh    Satvik Dixit    Rita Singh    Bhiksha Raj**
Carnegie Mellon University
{sdeshmuk, satvikd, rsingh, bhikshar}@andrew.cmu.edu

## Abstract

Multimodal Audio-Language Models (ALMs) can understand and reason over both audio and text. Typically, reasoning performance correlates with model size, with the best results achieved by models exceeding 8 billion parameters. However, no prior work has explored enabling small audio-language models to perform reasoning tasks, despite the potential applications for edge devices. To address this gap, we introduce Mellow, a small Audio-Language Model specifically designed for reasoning. Mellow achieves state-of-the-art performance among existing small audio-language models and surpasses several larger models in reasoning capabilities. For instance, Mellow scores 52.11 on MMAU, comparable to SoTA Qwen2 Audio (which scores 52.5) while using 50 times fewer parameters and being trained on 60 times less data (audio hrs). To train Mellow, we introduce ReasonAQA, a dataset designed to enhance audio-grounded reasoning in models. It consists of a mixture of existing datasets (30% of the data) and synthetically generated data (70%). The synthetic dataset is derived from audio captioning datasets, where Large Language Models (LLMs) generate detailed and multiple-choice questions focusing on audio events, objects, acoustic scenes, signal properties, semantics, and listener emotions. To evaluate Mellow's reasoning ability, we benchmark it on a diverse set of tasks, assessing on both in-distribution and out-of-distribution data, including audio understanding, deductive reasoning, and comparative reasoning. Finally, we conduct extensive ablation studies to explore the impact of projection layer choices, synthetic data generation methods, and language model pretraining on reasoning performance. Our training dataset, findings, and baseline pave the way for developing small ALMs capable of reasoning.

## 1   Introduction

In recent years, the field of computational audition [7] and machine listening has transitioned from task-specific approaches [13] to unified models [26, 14] capable of performing multiple tasks. These unified systems, known as Audio Foundation Models [69], have demonstrated impressive performance across a variety of tasks, often surpassing task-specific and dataset-specific models. Most Audio Foundation Models leverage language as a mode of supervision during pretraining, utilizing relatively simple objectives such as contrastive learning [25, 27, 73] or next-token prediction [14, 36, 33]. The simplicity of these objectives makes them highly scalable, allowing for the scaling of both data and model parameters with increased computational resources.

Over the years, advancements in architectural design, along with the scaling of both data and model parameters, have significantly enhanced performance on existing benchmarks [36, 34, 66, 33, 10, 11]. Early contrastive audio-language models, such as CLAP [25], were trained on relatively small datasets, such as 128k audio-text pairs. In contrast, recent next-token prediction models leverage datasets containing millions of pairs, incorporating multiple encoders and projection layers, with some models exceeding 10 million pairs [34]. Beyond benchmark improvements, these larger models have begun to exhibit emergent abilities [16, 18]—skills not explicitly trained for and not easily inferred from their individual components. Notable examples include the ability of Audio-Language Models (ALMs) to serve as general-purpose audio quality estimators [16], enhanced reasoning over

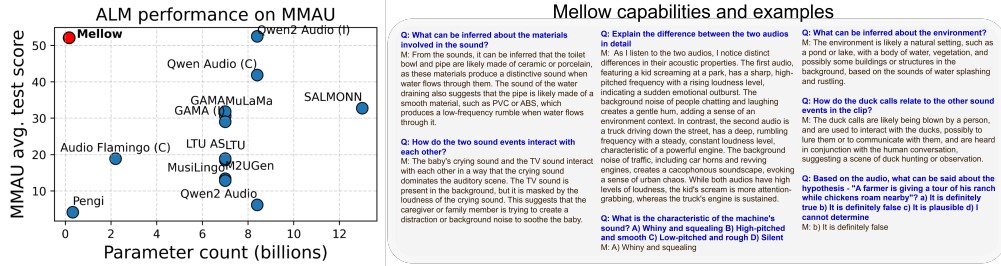

Figure 1: The left plot shows the performance of different ALMs on MMAU vs their parameter size. We plot models whose parameter counts are known, (I) indicates Instruction-tuned, (C) indicates Chat. The right figure shows Mellow's different capabilities and examples, the full examples are available in Figure 11.

audio-text pairs [34], and logical reasoning, particularly deductive reasoning [18]. Among these capabilities, the ability to perceive audio, reason over audio and text, and generate informed responses using world knowledge is particularly promising, with broad practical applications. This reasoning ability not only allows ALMs to tackle novel tasks but also provides a unified means to evaluate a core skill—one whose improvement leads to performance gains across multiple downstream applications.

Understanding and reasoning are related but distinct processes. Literature [42] defines understanding as grasping meaning and coherence, while reasoning involves drawing inferences. Similarly, [28] highlights that reasoning relies on distinct processing modes, reinforcing its distinction from general understanding. In audio models, reasoning can improve through two approaches. First, expanding data (audio) coverage (e.g., knowledge of audio objects and scenes), which provides a broader foundation upon which the model can apply its existing reasoning capabilities. Second, enhancing the model's inherent reasoning ability through architectural, learning or post-training techniques. Prior work [14, 36, 34, 33, 66, 10, 11, 18] has primarily followed the first approach, where increasing audio coverage indirectly enhances both understanding and reasoning performance. While effective, this approach does not improve the inherent reasoning ability constrained by architectural, learning, post-training techniques, or reasoning chains found in language not audio data. By fixing training audio, one can conduct controlled evaluations of ALMs, which can later scale via data expansion. We explore the second approach, aiming to enhance reasoning independently of data expansion.

In this paper, we introduce Mellow, a small Audio Language Model (ALM) that can reason over text and audio. The paper's main contributions are:

- **Mellow: small Audio-Language Model for reasoning.** We introduce Mellow, a small-scale ALM optimized for reasoning. Mellow surpasses all models within its parameter category and, in certain benchmarks, outperforms models with 50 times more parameters, while trained on only ∼152 hrs of audio. We provide model checkpoints, training data, and evaluation code to facilitate further research.

- **ReasonAQA: a training dataset for audio models.** We create ReasonAQA, a dataset designed to enhance the logical reasoning capabilities of ALMs. The dataset comprises a mixture of generated data (70%) and existing deductive and comparative reasoning datasets (30%) from the literature. To generate synthetic data, we leverage audio descriptions from AudioCaps and Clotho, using LLMs to create QA pairs focused on audio events, objects, acoustic scenes, acoustic properties, signal characteristics, semantics, and listener emotions. In total, ReasonAQA uses 56k (∼152 hrs) audio files and consists of 1M AQA instances, with predefined train, validation, and test splits.

- **Ablations and findings.** We start with a minimal audio-language model from the literature [14], which has previously been shown to perform at random on audio reasoning tasks [18, 63]. Then we conduct extensive ablation studies to analyze the impact of architectural choices, synthetic data generation, and training strategies on reasoning performance. We highlight the key findings in Section 5.

## 2 ReasonAQA

We construct the ReasonAQA dataset to train Audio-Language Models (ALMs) for reasoning over audio and text. This dataset is derived from a fixed set of audio files, allowing the study of improvements driven by architectural choices, synthetic data generation, and learning methodologies,

rather than those resulting from scale-related factors such as larger datasets, broader domains, or an increased number of learned concepts. The dataset construction and composition are detailed in Section 2.1 and Section 2.2, respectively.

## 2.1 Data construction

**Audio sources.** We use two audio datasets with human-labeled descriptions—AudioCaps [43] and Clotho [23]. These datasets are sourced from AudioSet and Freesound, respectively, and cover a wide range of audio events, acoustic scenes, and audio concepts. By restricting our audio sources for training and evaluation, we ensure consistency in audio concepts and isolate performance improvements resulting from model design and learning methods, rather than those driven by data scaling. AudioCaps annotations are generated using Amazon Mechanical Turk (MTurk), where annotators have access to both visual and audio content. This multimodal access allows for more detailed descriptions compared to those generated from audio-only annotations. In contrast, Clotho restricts annotators to audio-only listening, without access to visual or textual cues. Unlike AudioCaps, where audio clips are limited to 10 seconds, Clotho contains recordings ranging from 15 to 30 seconds, with each audio file accompanied by five descriptions, each containing 8 to 20 words. These datasets have been widely used for building and benchmarking audio captioning models [53, 54, 55, 56, 17] and text-to-audio retrieval models [45]. Additionally, they have been extended to construct datasets for audio question-answering [48], deductive reasoning [18], and comparative reasoning [19]. Given their diversity and extensive utilization across multiple tasks, we select these two datasets as the foundation for generating data in ReasonAQA.

**Audio reasoning tasks.** In the literature, audio reasoning tasks utilizing AudioCaps and Clotho primarily include audio entailment, a classification task, and audio difference explanation, a description task, in addition to the traditional audio captioning task. To incorporate these tasks into ReasonAQA, we convert both audio captioning and audio entailment into question-answering (QA) formats. Specifically, audio captioning is transformed into a descriptive question-answering task, while audio entailment is converted into a multiple-choice question (MCQ) format. The audio difference explanation task is already QA-based, so we integrate it directly into ReasonAQA.

**ReasonAQA dataset (1.24M)**

| Dataset | Type | | | Audio Source | | Train | | | Val | | | Test | | |
|---|---|---|---|---|---|---|---|---|---|---|---|---|---|---|
| | Binary | MCQ | Desc. | AudioCaps | Clotho | # Audio | # QA Pairs | Per % | # Audio | # QA Pairs | Per % | # Audio | # QA Pairs | Per % |
| Clotho [23] | | | ✓ | | ✓ | 3839 | 19195 | 2.0% | 1045 | 5225 | 4.7% | 1045 | 5225 | 3.2% |
| AudioCaps [43] | | | ✓ | ✓ | | 48660 | 48660 | 5.0% | 494 | 494 | 0.4% | 957 | 4785 | 3.0% |
| **Sum** | | | | | | | 67855 | 7.0% | | 5719 | 5.1% | | 10010 | 6.2% |
| ClothoAQA [48] | ✓ | | | | ✓ | | 14088 | 1.5% | | 4128 | 3.7% | | 5676 | 3.5% |
| CLE [18] | | ✓ | | | ✓ | | 11511 | 1.2% | | 3132 | 2.8% | | 3135 | 1.9% |
| ACE [18] | | ✓ | | ✓ | | | - | - | - | - | - | | 14355 | 8.9% |
| CLD [19] | | | ✓ | | ✓ | | 57585 | 5.9 % | | 15675 | 14.0% | | 15675 | 9.7% |
| ACD [19] | | | ✓ | ✓ | | | 145980 | 15.1% | | 7368 | 6.6% | | 14040 | 8.7% |
| **Sum** | | | | | | | 229164 | 23.7% | | 30303 | 27.0% | | 52881 | 32.7% |
| Clotho-MCQ | | ✓ | | | ✓ | | 95551 | 9.9% | | 26054 | 23.2% | | 26031 | 16.1% |
| Clotho-Detail | | | ✓ | | ✓ | | 94838 | 9.8% | | 25785 | 23.0% | | 25851 | 16.0% |
| AudioCaps-MCQ | | ✓ | | ✓ | | | 241531 | 24.9% | | 12236 | 10.9% | | 23672 | 14.6% |
| AudioCaps-Detail | | | ✓ | ✓ | | | 239132 | 24.7% | | 12110 | 10.8% | | 23355 | 14.4% |
| **Sum** | | | | | | | 671052 | 69.3% | | 76185 | 67.9% | | 98909 | 61.1% |
| **Total** | ✓ | ✓ | ✓ | ✓ | ✓ | 52499 | 968071 | 100% | 1539 | 112207 | 100% | 2002 | 161800 | 100% |

Table 1: The composition of the `ReasonAQA` dataset. The training set is restricted to AudioCaps and Clotho audio files and the testing is performed on 6 tasks - Audio Entailment, Audio Difference, ClothoAQA, Clotho MCQ, Clotho Detail, AudioCaps MCQ and AudioCaps Detail.

**Synthetic data.** We use audio descriptions from AudioCaps and Clotho to prompt an LLM for generating question-answer (QA) pairs. This method of LLM prompting has previously been used to create various datasets and benchmarks [57, 36, 18, 19]. Following this approach, we generate synthetic data with two key distinctions. First, we produce a mix of detailed and multiple-choice (MCQ) questions, unlike OpenAQA, which contains only descriptive questions. Second, the questions are explicitly grounded in audio, setting them apart from existing datasets [36], which is shown in the failure modes of existing datasets (Table 12). We refer the reader to Appendix C for details on

the data generation methodology, prompting setup, data analysis, and a comparative evaluation with existing datasets, including OpenAQA.

## 2.2 Data composition

Using AudioCaps and Clotho audio files, along with existing audio tasks and synthetic data, we construct the train, validation, and test subsets of ReasonAQA. The dataset composition for each split is shown in Table 1. Each example in the dataset is structured as a quadruple: `audio1`, `audio2`, `question`, `answer`, where `audio`$_2$ is empty when the task does not require two audio inputs.

## 3 Mellow

In this section, we introduce Mellow, a small Audio Language Model that processes two audios inputs and text prompts to generate text-based outputs. The architecture of Mellow is illustrated in Fig. 2.

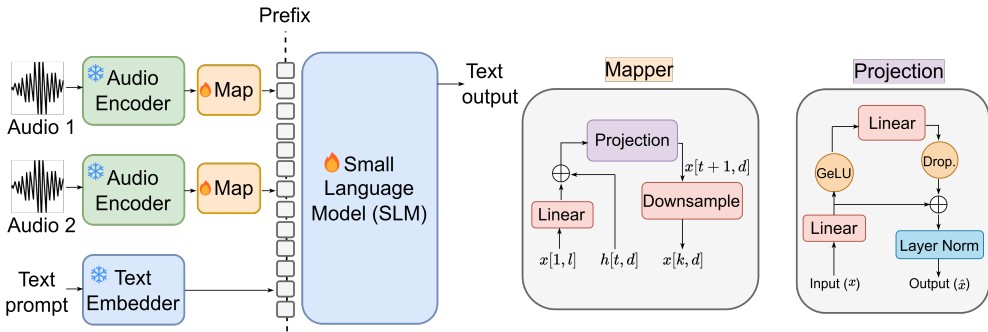

Figure 2: Mellow takes two audio recordings and a text prompt as input and generates a text output. The two audio inputs are encoded by an audio encoder and projected into the language model space using a mapper (map). Simultaneously, the text prompt is embedded by the text embedder. The audio projection 1, audio projection 2, and text embedding are concatenated to form the prefix. During concatenation, a separator token ($s$) is inserted between the three components (as shown in Fig. 2). The prefix is then used to prompt the small Language Model, which generates a natural language response.

**Audio Encoder.** The raw audio with its high sampling rate is unsuitable for the Language Model to operate on. Therefore, we encode the raw audio using an audio encoder, which produces a fixed-length latent representation. We use HTSAT [9], a Hierarchical Token Semantic Transformer, as the default audio encoder. HTSAT, based on the Swin Transformer, has demonstrated state-of-the-art performance on AudioSet [31]. The audio encoder output consists of a latent prediction, which is then expanded by a token-semantic CNN to generate an event presence map with temporal information. The audio encoder outputs passed to the mapper are: (1) the latent prediction ($1 \times l$), which acts as a CLS token, and (2) the event presence map (time frames $\times$ d), which retains temporal information across the audio. The encoder processes 10-second audio clips sampled at 32 kHz.

**Mapper.** The audio encoder output is projected into the language model space using a mapper. The mapper consists of a linear layer, projection layer, and downsampler. The linear layer projects the latent prediction ($1 \times l$), which acts as a CLS token into the same space as the event presence map (time frames $\times$ d), which retains temporal information across the audio. After the linear layer projection, the two outputs are concatenated and passed to projection layer. Existing literature explores various projection mechanisms [26, 27, 14, 33], ranging from simple linear transformations to full transformer-based mappers. We use a non-linear projection consisting of two linear layers whose outputs are combined before passing through a layer normalization. Finally, the output of the projection layer is downsampled before latent prompting of Language Model. The details of the mapper and projection layers, along with ablation studies, are provided in Appendix E and Section 5.

**Language Model.** We specifically choose small Language Models (SLMs) over large or traditional Language Models, aiming to develop a minimal yet effective recipe for small Audio-Language Models that can reason. For this purpose, we select smolLM2 [2], one of the best-performing SLMs on various benchmarks [12, 6, 41, 38, 62, 74, 65, 58]. We conduct ablation studies to evaluate the

impact of language model pretraining and architecture on audio reasoning performance. These results are presented in Section 5 and Table 14.

**Training.** Mellow is trained on the next-token prediction task, where the next-token is predicted based on past-tokens and the two input audios ($A_1$, $A_2$). For each token, cross-entropy ($P$) is computed between predicted probability and ground-truth, which is then summed over tokens. The objective is:

$$P(x_t \mid x_{1:t-1}, A_1, A_2, T) \tag{1}$$

using cross-entropy for all $1 < t \leq T$, it is conditioned on the text sequence $x_{1:T}$ along with the input audios $A_1$ and $A_2$.

**Inference.** The literature uses different inference or sampling techniques ranging from greedy decoding, to top-k and top-p [36, 34, 19], beam search [14], etc. Each technique offers advantages such as task-specific benefits, increased diversity, and improved performance. For all evaluations in this work, we employ top-p sampling where p is set to 0.8 and a temperature of 1.

## 4 Results

In this section, we evaluate Mellow across various audio reasoning tasks. Specifically, we compare Mellow against other Audio-Language Models on the following benchmarks: MMAU (Section 4.1), Audio Entailment (Section 4.2), Audio Difference (Section 4.3), Audio Captioning and Binary AQA (Section 4.4). The multiple tasks benchmark different types of reasoning ability of Mellow.

### 4.1 Understanding and reasoning

In this section, we evaluate Mellow on its multimodal audio understanding and reasoning capabilities. We use the MMAU benchmark [63], which consists of 10k annotated Audio Question-Answering (AQA) pairs spanning the speech, sound, and music domains. This benchmark assesses a model's ability across 27 distinct skills covering various reasoning tasks. Notably, Mellow is trained exclusively on AudioCaps and Clotho audio files, whereas MMAU sources its audio from 10 different datasets, with no overlap with Mellow's training data. This makes MMAU an ideal benchmark for evaluating Mellow's reasoning performance in an Out-of-Distribution (OOD) setting. Additionally, it allows us to assess how effectively our training data (ReasonAQA) enhances reasoning ability in ALMs while minimizing improvements driven by the scaling of audio concepts.

| Models | Size | {So, Mu, Sp} | Sound | | Music | | Speech | | Avg | |
|---|---|---|---|---|---|---|---|---|---|---|
| | | | Test-mini | Test | Test-mini | Test | Test-mini | Test | Test-mini | Test |
| Random Guess | - | - | 26.72 | 25.73 | 24.55 | 26.53 | 26.72 | 25.50 | 26.00 | 25.92 |
| Most Frequent Choice | - | - | 27.02 | 25.73 | 20.35 | 23.73 | 29.12 | 30.33 | 25.50 | 26.50 |
| Human (test-mini) | - | - | 86.31 | - | 78.22 | - | 82.17 | - | 82.23 | - |
| **Large Audio Language Models (LALMs)** | | | | | | | | | | |
| LTU | 7B | ✓ ✓ ✗ | 22.52 | 25.86 | 09.69 | 12.83 | 17.71 | 16.37 | 16.89 | 18.51 |
| LTU AS | 7B | ✓ ✓ ✓ | 23.35 | 24.96 | 9.10 | 10.46 | 20.60 | 21.30 | 17.68 | 18.90 |
| MusiLingo | 7B | ✗ ✓ ✗ | 23.12 | 27.76 | 03.96 | 06.00 | 05.88 | 06.42 | 10.98 | 13.39 |
| MuLLaMa | 7B | ✗ ✓ ✗ | 40.84 | 44.80 | 32.63 | 30.63 | 22.22 | 16.56 | 31.90 | 30.66 |
| M2UGen | 7B | ✗ ✓ ✗ | 03.60 | 03.69 | 32.93 | 30.40 | 06.36 | 04.53 | 14.28 | 12.87 |
| GAMA | 7B | ✓ ✓ ✗ | 41.44 | 45.40 | 32.33 | 30.83 | 18.91 | 19.21 | 30.90 | 31.81 |
| GAMA-IT | 7B | ✓ ✓ ✗ | 43.24 | 43.23 | 28.44 | 28.00 | 18.91 | 15.84 | 30.20 | 29.02 |
| Qwen-Audio-Chat | 8.4B | ✓ ✗ ✗ | 55.25 | **56.73** | 44.00 | 40.90 | 30.03 | 27.95 | 43.10 | 41.86 |
| Qwen2-Audio | 8.4B | ✓ ✓ ✓ | 07.50 | 08.20 | 05.14 | 06.16 | 03.10 | 04.24 | 05.24 | 06.20 |
| Qwen2-Audio-Instruct | 8.4B | ✓ ✓ ✓ | 54.95 | 45.90 | **50.98** | **53.26** | 42.04 | 45.90 | 49.20 | 52.50 |
| SALAMONN | 13B | ✓ ✓ ✓ | 41.00 | 40.30 | 34.80 | 33.76 | 25.50 | 24.24 | 33.70 | 32.77 |
| Gemini Pro v1.5 | - | - | **56.75** | 54.46 | 49.40 | 48.56 | **58.55** | **55.90** | **54.90** | **52.97** |
| **Large Language Models (LLMs)** | | | | | | | | | | |
| GPT4o + weak cap. | - | - | 39.33 | 35.80 | 39.52 | 41.9 | 58.25 | 68.27 | 45.70 | 48.65 |
| GPT4o + strong cap. | - | - | **57.35** | **55.83** | 49.70 | **51.73** | 64.86 | 68.66 | **57.30** | **58.74** |
| Llama-3-Ins. + weak cap. | 8B | - | 34.23 | 33.73 | 38.02 | 42.36 | 54.05 | 61.54 | 42.10 | 45.87 |
| Llama-3-Ins. + strong cap. | 8B | - | 50.75 | 49.10 | **50.29** | 48.93 | 55.25 | 62.70 | 52.10 | 53.57 |
| **Small Audio Language Models (SALMs)** | | | | | | | | | | |
| Pengi | 323M | ✓ ✓ ✗ | 06.10 | 08.00 | 02.90 | 03.05 | 01.20 | 01.50 | 03.40 | 04.18 |
| Audio Flamingo Chat | 2.2B | ✓ ✓ ✗ | 23.42 | 28.26 | 15.26 | 18.20 | 11.41 | 10.16 | 16.69 | 18.87 |
| **Mellow** | 167M | ✓ ✗ ✗ | **61.26** | **64.90** | **54.19** | **52.67** | **29.73** | **38.77** | **48.40** | **52.11** |

Table 2: Results on MMAU benchmark. The third column indicates the training data used to train these models containing sound, music, and speech. The best-performing models in each category are highlighted in **bold**, and the second-best scores are underlined.

**Results.** The results are presented in Table 2. We compare Mellow against three types of models: Large Audio Language Models (LALMs), Large Language Models (LLM) and Small Audio Language Models (SALMs). For LALMs and SALMs, the models take audio and text questions as input and produce text responses as output. In contrast, LLMs do not support audio input. To accommodate this, audio captions using a LALM [11] are fed to the LLM. The strong and weak captions refer to different captioning models and prompting setups; for details, we refer readers to the MMAU paper [63]. Analyzing the results, we see the following trends. First, Mellow achieves state-of-the-art (SoTA) performance in reasoning over sound and music, outperforming all existing SALMs, LALMs, and LLMs. Second, Mellow is trained on only 51k unique audio files and has 167M parameters. In contrast, LALMs utilize 100× more audio data and 100× more parameters to achieve similar reasoning capabilities. This makes Mellow a highly competitive option for on-device audio understanding and reasoning. Third, Mellow performs poorly on speech-based tasks since it is not trained on any speech content (e.g., ASR-based datasets). However, it still learns attributes such as gender and music emotion, achieving a score of 29, which is the best among SALMs and above random performance.

**LM reliance.** Historically, Audio Question-Answering (AQA) models have exhibited a heavy reliance on the Language Model (LM), often to the extent of overlooking audio input while still achieving strong performance [48, 63]. To investigate this issue, we conduct an experiment similar to those in ClothoAQA [48] and MMAU [63], where we replace the audio input with Gaussian noise and evaluate the model's performance. The results of this experiment are presented in Fig. 3. We observe that Mellow's performance drops by approximately 13% when Gaussian noise is used instead of actual audio. Compared to Mellow, Gemini 1.5 Pro and Qwen2 Instruct exhibit

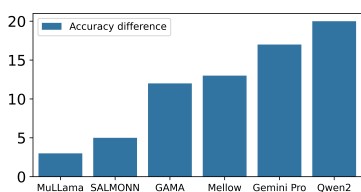

Figure 3: Absolute difference in accuracy between performance with real audio input and performance with Gaussian noise.

stronger grounding in audio, which can be attributed to two key factors: (1) Speech-based tasks constitute nearly 30% of MMAU, and Mellow produces near-random responses on these tasks due to limited speech training, rarely conditioning its outputs on the audio input. (2) Models like Gemini Pro are natively multimodal, meaning they share tokenizers for both audio and text, whereas Mellow is an LM conditioned on audio rather than a fully integrated multimodal model. Overall, there is a need to reduce the perception gap in Mellow [18, 32], as improving its ability to incorporate audio cues effectively can significantly enhance its audio reasoning capabilities.

**Performance distribution.** We conduct an error analysis of Mellow on MMAU, specifically examining its performance across question difficulty levels and task categories. MMAU categorizes question difficulty into easy, medium, and hard, while tasks are divided into 27 distinct skills. Table 3 presents Mellow's performance across different difficulty levels. Notably, Mellow performs better on medium-difficulty questions compared to easy and hard ones, suggesting that it has stronger reasoning capabilities but weaker general understanding. Across different task

| Model | Easy | Medium | Hard |
|---|---|---|---|
| SALMONN | 20.31 | 39.33 | 30.63 |
| GAMA | 31.36 | 35.70 | 22.85 |
| Qwen2 | 50.59 | 55.63 | 46.99 |
| Gemini Pro v.5 | **57.04** | 51.49 | **52.07** |
| Mellow | 50.68 | **60.10** | 32.89 |

Table 3: Model performance across difficulty levels on MMAU

categories, Mellow performs best on: acoustic source inference, ambient sound interpretation, and music texture interpretation. However, it struggles with phoneme stress pattern analysis, emotion flip detection, and lyrical reasoning. These results indicate that Mellow performs poorly on speech-related tasks, while its performance on sound and music tasks is relatively stronger.

## 4.2 Deductive reasoning

In this section, we evaluate Mellow's deductive reasoning ability across both audio and text using the Audio Entailment benchmark [18]. In this benchmark, each example consists of audio ($\mathcal{A}$) and text hypothesis ($\mathcal{H}$). The task is to determine whether the hypothesis is true given the audio input. The answer falls into one of three categories: Entailment ($\mathcal{H}$ is true given $\mathcal{A}$), Neutral ($\mathcal{H}$ is plausible but not necessarily true given $\mathcal{A}$), or Contradiction ($\mathcal{H}$ is false given $\mathcal{A}$). Beyond assessing deductive reasoning, this task also helps identify audio hallucinations, which typically appear in two forms. The first is inferred cues, where the model introduces audio concepts that are not present in the input. The second is contextual assumptions, where the model interprets sounds based on text likelihood rather

than actual audio evidence. This evaluation provides insight into Mellow's ability to reason logically over multimodal inputs while also highlighting potential failure modes related to audio hallucination.

| Models | LLM (param) | CLE | | | | | | | ACE | | | | | | |
|---|---|---|---|---|---|---|---|---|---|---|---|---|---|---|---|
| | | ACC | P | R | F1 | EACC | NACC | CACC | ACC | P | R | F1 | EACC | NACC | CACC |
| CLAP 22 | 110M | 45.90 | 54.99 | 45.90 | 46.56 | 60.00 | 40.29 | 37.42 | 44.34 | 44.35 | 43.34 | 43.32 | 43.32 | 56.41 | 45.08 |
| LCLAP | 125M | 51.13 | 55.44 | 51.13 | 51.61 | 66.79 | 36.46 | 50.14 | 58.72 | 57.67 | 58.72 | 56.93 | 28.67 | 59.00 | 88.48 |
| CLAP 23 | 124M | 51.64 | 51.55 | 51.63 | 51.59 | 41.53 | 40.38 | 73.01 | 48.60 | 46.78 | 48.60 | 46.56 | 48.80 | 20.02 | 76.99 |
| Pengi-noenc | 124M | 27.81 | 18.43 | 27.81 | 22.16 | 49.67 | 0.00 | 33.78 | 26.29 | 16.99 | 26.29 | 20.45 | 53.12 | 0.00 | 25.75 |
| Pengi-enc | 124M | 37.26 | 24.65 | 37.26 | 28.88 | 75.41 | 0.00 | 36.36 | 38.67 | 25.58 | 38.67 | 30.39 | 73.35 | 0.00 | 42.65 |
| LTU-AS | 7B | 36.81 | 37.37 | 36.81 | 34.20 | 62.78 | 31.87 | 15.79 | 36.33 | 37.72 | 36.33 | 33.34 | 67.02 | 24.35 | 17.62 |
| Qwen-A | 7B | 36.20 | 40.12 | 36.20 | 31.17 | 76.75 | 13.88 | 17.99 | 35.63 | 35.62 | 35.63 | 32.19 | 66.69 | 13.23 | 26.96 |
| Qwen-AC | 7B | 54.42 | 56.04 | 54.42 | 49.75 | 90.24 | 15.69 | 57.32 | 52.16 | 56.69 | 52.16 | 49.18 | 93.00 | 28.21 | 35.28 |
| GAMA | 7B | 48.26 | 61.51 | 48.26 | 45.34 | 81.44 | 41.24 | 22.11 | 52.48 | 65.31 | 52.48 | 49.33 | 78.27 | 58.85 | 20.31 |
| GAMA-IT | 7B | 39.74 | 56.04 | 39.74 | 34.33 | 79.23 | 29.47 | 10.53 | 41.67 | 56.72 | 41.67 | 38.28 | 78.52 | 26.96 | 19.54 |
| SALMONN | 13B | 52.22 | 50.54 | 52.22 | 45.15 | 67.75 | 7.08 | 81.82 | 56.22 | 55.51 | 56.22 | 48.26 | 71.14 | 6.98 | 90.55 |
| **Mellow** | 135M | **91.16** | **91.35** | **91.16** | **91.10** | **90.53** | **85.26** | **97.70** | **89.66** | **90.71** | **89.66** | **89.34** | **95.82** | **73.48** | **99.67** |

Table 4: Deductive reasoning ability of different ALMs. The evaluation is performed on Audio Entailment task

**Results.** The results on the CLE and ACE datasets for audio entailment are presented in Table 4. The first three entries correspond to contrastive Audio-Language Models (ALMs), which are naturally suited for classification tasks, while the remaining entries represent next-token prediction ALMs. Mellow outperforms all existing ALMs, regardless of whether they use small or large language models and whether they are contrastive or next-token prediction models. Mellow excels at identifying hypotheses that are definitely true or definitely false given the audio but struggles with detecting plausible scenarios. For instance, on the ACE dataset, Mellow correctly classifies plausible (NACC) hypotheses only 75% of the time, whereas its accuracy for entailment and contradiction cases is in the high 90s. Despite this limitation, Mellow still outperforms the similarly sized next-token prediction ALM, Pengi [14], which fails to detect plausible scenarios effectively.

**Linear-probe.** Zero-shot evaluation is not a fair comparison, as Mellow has been explicitly trained for deductive reasoning. To provide a more balanced assessment, we compare Mellow against linear-probe (supervised) performance. In this setup, the audio encoder extracts the audio embeddings, while the text encoder extracts the text embeddings from the hypothesis. These embeddings are then fed into a linear classifier, which is trained to predict one of three entailment categories: entailment, neutral, or contradiction. The classifier is trained on the training split

| Models | ACC | P | R | F1 |
|---|---|---|---|---|
| CLAP 22 | 71.10 | 71.30 | 71.10 | 71.18 |
| LCLAP | 74.35 | 74.70 | 74.35 | 74.45 |
| CLAP 23 | 83.29 | 83.61 | 83.29 | 83.36 |
| Pengi-enc | 76.27 | 76.74 | 76.27 | 76.42 |
| CLAP 23* | 86.40 | 86.71 | 86.40 | 86.47 |
| **Mellow** | **91.16** | **91.35** | **91.16** | **91.10** |

Table 5: Supervised performance on CLE dataset of Audio Entailment task

and evaluated on the test split of the CLE dataset. In Table 5, the * symbol indicates the "caption-before-reasoning" method, which incorporates an additional audio captioning step before reasoning. This approach has been shown to improve performance [18, 32]. Overall, even in the supervised setup, Mellow outperforms all other ALMs, demonstrating its strong deductive reasoning capabilities.

## 4.3 Comparative reasoning

In this section, we evaluate Mellow's comparative reasoning ability (reasoning by analogy) over two audio inputs and text. We use the Audio Difference benchmark [19], where each example consists of two audio recordings and a text question, with the goal of identifying the differences between the two audios. The text question specifies the level of detail required in the answer and is categorized into three tiers: Tier 1 (concise), Tier 2 (brief), and Tier 3 (detailed). Beyond assessing comparative reasoning, this task requires the model to integrate audio information with world knowledge to effectively distinguish between the two audios. It demands an understanding of signal properties (e.g., frequency, amplitude) and contextual cues (e.g., genre, environment) to identify both differences and similarities, making it a strong benchmark for evaluating a model's ability to combine audio perception with world knowledge.

**Results.** Table 6 presents the results on the Audio Difference (ACD and CLD) datasets. We compare Mellow with various ALMs on this benchmark. All models are trained on the ACD and CLD training sets and evaluated on their respective test sets. Among large Audio Language Models, we have QwenAC, where the labels (L) and (F) for QwenAC indicate LoRA and full finetuning on the training

| Models | LLM (param) | CLD-1 | | CLD-2 | | CLD-3 | | ACD-1 | | ACD-2 | | ACD-3 | |
|---|---|---|---|---|---|---|---|---|---|---|---|---|---|
| | | BLEU$_4$ | SPICE | BLEU$_4$ | SPICE | BLEU$_4$ | SPICE | BLEU$_4$ | SPICE | BLEU$_4$ | SPICE | BLEU$_4$ | SPICE |
| Baseline | 125M | 8.8 | 6.4 | 26.5 | 26.4 | 13.7 | 17.2 | 13.1 | 10.1 | 21.7 | 21.5 | 15.1 | 13.8 |
| QwenAC (L) | 7B | 9.3 | 8.2 | 26.4 | 21.1 | 13.5 | 14.5 | 14.5 | 9.5 | 22.0 | **24.0** | 16.1 | 16.0 |
| QwenAC (F) | 7B | 7.6 | 9.5 | **28.5** | **27.3** | 12.2 | 14.9 | 12.3 | 9.4 | 21.7 | 21.9 | 14.5 | 14.3 |
| ADIFF | 125M | 15.3 | 11.9 | 24.5 | 23.2 | 17.1 | 16.7 | 14.8 | 12.7 | 23.4 | 22.2 | 16.9 | 17.1 |
| **Mellow** | 135M | **17.3** | **13.9** | 26.1 | 25.0 | **17.9** | **17.3** | **15.6** | **13.9** | **24.3** | 23.7 | **17.9** | **18.7** |

Table 6: Comparative reasoning ability of different ALMs. The evaluation is performed on the Audio Difference Explanation task on datasets CLD and ACD. The sampling setup is kept constant across models (Appendix E.3).

data, respectively. Among smaller ALMs, we have a naive baseline built on a Pengi-like architecture [14] and ADIFF [19], which improves upon Pengi and achieves state-of-the-art results on this task. From these results, we see that Mellow consistently outperforms existing ALMs on Tier-1 and Tier-3. On Tier-2, however, QwenAC outperforms Mellow in terms of BLEU$_4$ on both CLD and ACD. Linguistically, Tier-1 is the hardest to learn, followed by Tier-3, with Tier-2 being the easiest. In particular, Tier-2 contains roughly 15% of words related to linguistics and stop words rather than specific audio-contrasting information, whereas Tier-1 contains fewer words but primarily focuses on audio details. Consequently, Tier-2's higher scores, even in subsequent experiments, can be attributed to its linguistic simplicity. As a result, Mellow's smaller LM falls short of the larger QwenAC LM in producing more coherent language structures on Tier-2.

## 4.4 Audio captioning and binary AQA

In this section, we evaluate Mellow on audio captioning and Audio Question-Answering (AQA) tasks. Higher performance in audio captioning indicates a smaller perception gap and better grounding in audio for ALMs [18, 32]. Traditionally, ALMs have also been benchmarked on binary question-answering, which involves yes/no questions covering count-based and evidence-based reasoning. Therefore, we assess Mellow's performance on both Audio Captioning and the ClothoAQA.

**Results.** The results are presented in Table 7 Mellow outperforms the existing small ALM, Pengi, across all three evaluation metrics. Specifically, Mellow achieves a SPICE score of 17.8 for audio captioning, significantly higher than Pengi's 12.7, and an accuracy of 71.4% on ClothoAQA, surpassing Pengi's 63.6%. These results suggest that Mellow is particularly effective at generating semantically meaningful captions for audio events and performing binary question-answering tasks. Compared to larger models like Qwen-Audio and GAMA, Mellow achieves competitive performance on

| Models | LLM (param) | Audio Caption | | AQA |
|---|---|---|---|---|
| | | AC (SPICE) | CL (SPICE) | CL (ACC) |
| *Large Audio Language Models (LALMs)* | | | | |
| LTU | 7B | 16.9 | 11.7 | 25.1 |
| SALMONN | 13B | 8.3 | 7.6 | 23.1 |
| AudioGPT | - | 6.9 | 6.2 | 33.4 |
| GAMA | 7B | **18.5** | **13.5** | **71.6** |
| QwenAC | 7B | 14.7 | 9.8 | 32.3 |
| *Small Audio Language Models (SALMs)* | | | | |
| Pengi | 124M | 12.7 | 7.0 | 63.6 |
| **Mellow** | 135M | **17.8** | **9.4** | **71.4** |

Table 7: Captioning and AQA performance of ALMs.

binary AQA while using just 2.4% of their parameter size. However, on captioning, we see Mellow falling short of LALMs, where large audio concept knowledge is necessary to produce better captions.

## 5 Ablation findings

We perform ablation studies to identify which components most effectively enhance reasoning in audio-language models, with a particular focus on Small Audio-Language Models. In particular, we examine the effects of audio encoders (Section F.2), language model choice (Section F.3), projection layers (Section F.1), prefix-tuning versus fine-tuning (Section F.4), LoRA adaptation (Section F.5), synthetic data generation (Section F.6), and scaling data (Section F.7). The results are shown in Table 8 and the detailed experimental setups and analyses are available in Appendix F. The key observations from these ablation studies are:

**Fine-tuning outperforms prefix-tuning.** Prefix-tuning and LoRa adaptation underperform relative to full fine-tuning, even when larger transformer-based mappers are used. In contrast, fine-tuning the language model with a small linear or two-layer non-linear mapper offers a better balance between computational cost and performance. For LoRA ablation and settings, please refer Appendix F.5.

**Better LM pretraining improves ALM reasoning on open-ended tasks.** Replacing GPT-2 with SmolLM2, while keeping the data and architecture unchanged, results in higher performance on

| Models | Size | Audio Caption | | B-AQA | Audio Entailment | | Audio Difference | | MMAU (test-mini) | | | |
|---|---|---|---|---|---|---|---|---|---|---|---|---|
| | | AC (SPICE) | CL (SPICE) | ClothoAQA (ACC) | CLE (ACC) | ACE (ACC) | CLD-3 (SPICE) | ACD-3 (SPICE) | Sound (ACC) | Music (ACC) | Speech (ACC) | Avg. (ACC) |
| *The SLM is GPT2 and frozen and the projection layer is changed* | | | | | | | | | | | | |
| Linear | 156M | 4.53 | 5.98 | 53.23 | 30.79 | 30.15 | 5.21 | 8.39 | 16.80 | 31.25 | 30.10 | 26.05 |
| Non-linear 2 | 157M | 4.79 | 6.10 | 55.18 | 32.12 | 29.56 | 5.56 | 8.62 | 17.42 | 33.23 | 27.03 | 25.89 |
| Transformer [14] | 195M | 10.26 | 7.89 | 62.35 | 43.89 | 42.10 | 11.88 | 12.50 | 28.65 | 35.23 | 26.68 | 30.19 |
| *The SLM is GPT2 and finetuned and the projection layer is changed* | | | | | | | | | | | | |
| Linear | 156M | 9.87 | 7.10 | 70.90 | 93.45 | 92.90 | 13.65 | 14.21 | 47.64 | 45.39 | 27.00 | 40.01 |
| Non-linear 2 | 157M | 10.51 | 7.15 | 71.25 | 93.40 | 93.27 | 13.77 | 14.01 | 48.05 | 48.50 | 27.33 | 41.29 |
| Transformer [14] | 195M | 10.77 | 7.23 | 70.89 | 92.32 | 93.65 | 13.45 | 15.21 | 47.89 | 49.10 | 27.10 | 41.36 |
| *The SLM is frozen, projection layer is non-linear and the SLM is changed* | | | | | | | | | | | | |
| GPT2 frozen | 157M | 4.79 | 6.10 | 55.18 | 32.12 | 29.56 | 5.56 | 8.62 | 17.42 | 33.23 | 27.03 | 25.89 |
| SmolLM2 frozen | 167M | 5.26 | 8.58 | 45.70 | 33.40 | 31.66 | 11.43 | 13.62 | 35.14 | 30.54 | 20.12 | 28.60 |
| *The SLM is finetuned, projection layer is non-linear and the SLM is changed* | | | | | | | | | | | | |
| GPT2 finetune | 157M | 10.51 | 7.15 | 71.25 | 93.40 | 93.27 | 13.77 | 14.01 | 48.05 | 48.50 | 27.33 | 41.29 |
| SmolLM2 finetune | 167M | 18.60 | 9.83 | 71.65 | 92.00 | 90.85 | 17.33 | 18.68 | 59.46 | 50.60 | 28.82 | 46.29 |
| *Diferent finetuning methods. The SLM is SmolLM2, projection layer is non-linear* | | | | | | | | | | | | |
| Prefix-tuning | 167M | 5.26 | 8.58 | 45.70 | 33.40 | 31.66 | 11.43 | 13.62 | 35.14 | 30.54 | 20.12 | 28.60 |
| LoRA (8, 16) | 167M | 18.53 | 9.25 | 64.64 | 79.33 | 84.15 | 14.23 | 14.98 | 45.95 | 42.51 | 28.83 | 39.10 |
| LoRA (256, 512) | 181M | 19.01 | 10.59 | 65.54 | 86.66 | 89.87 | 15.36 | 16.51 | 50.75 | 49.10 | 33.33 | 44.40 |
| Finetuning | 167M | 18.60 | 9.83 | 71.65 | 92.00 | 90.85 | 17.33 | 18.68 | 59.46 | 50.60 | 28.82 | 46.29 |
| *The SLM is SmolLM2 and finetuned, projection layer is non-linear and the audio encoder is changed* | | | | | | | | | | | | |
| CNN14 | 219M | 15.97 | 7.91 | 65.82 | 91.06 | 92.39 | 16.27 | 16.95 | 54.05 | 47.60 | 28.23 | 43.30 |
| HTSAT | 167M | 18.60 | 9.83 | 71.65 | 92.00 | 90.85 | 17.33 | 18.68 | 59.46 | 50.60 | 28.82 | 46.29 |
| *The SLM is SmolLM2 and finetuned, projection layer is non-linear, and training data is changed* | | | | | | | | | | | | |
| Type 1 | 167M | 18.60 | 9.83 | 71.65 | 92.00 | 90.85 | 17.33 | 18.68 | 59.46 | 50.60 | 28.82 | 46.29 |
| Type 2 | 167M | 16.47 | 8.23 | 71.05 | 92.50 | 93.20 | 16.98 | 18.09 | 59.45 | 42.81 | 37.84 | 46.70 |
| Type 3 | 167M | 17.43 | 9.88 | 66.83 | 91.87 | 94.25 | 17.37 | 18.67 | 61.56 | 45.21 | 32.43 | 46.40 |
| Type 4 | 167M | 17.79 | 9.38 | 71.39 | 91.16 | 89.66 | 17.21 | 18.54 | 61.26 | 54.19 | 29.73 | 48.40 |
| *The SLM is SmolLM2 and finetuned, projection layer is non-linear, and WavCaps is added to training* | | | | | | | | | | | | |
| Type 1 | 167M | 18.60 | 9.83 | 71.65 | 92.00 | 90.85 | 17.33 | 18.68 | 59.46 | 50.60 | 28.82 | 46.29 |
| + WavCaps | 167M | 14.83 | 9.66 | 71.32 | 92.47 | 92.69 | 17.92 | 19.13 | 59.16 | 60.48 | 23.72 | 47.80 |
| **Mellow** | 167M | 17.79 | 9.38 | 71.39 | 91.16 | 89.66 | 17.21 | 18.54 | 61.26 | 54.19 | 29.73 | 48.40 |

Table 8: Ablation study results. The experimental setup for each experiment is described in Appendix F.

reasoning tasks. The benefits of improved pretraining are most apparent on open-ended reasoning, where the model must generate long, coherent responses grounded in audio. For deductive reasoning, performance is comparable and only shows improvement when a larger LM (>1B params) is used.

**Better unimodal audio representations improve LM reasoning via coverage.** Pretraining audio models on large audio corpora—through supervised or SSL approaches—is essential for learning fundamental audio concepts that an LLM can reason over. Enhanced audio representations, as shown by linear-probe evaluations or downstream task performance, directly lead to better reasoning by improving coverage. For example, HTSAT achieves higher mAP on AudioSet than CNN14, resulting in improved performance on MMAU and captioning. This improvement primarily stems from increased coverage and leveraging existing reasoning patterns rather than learning new ones.

**Reasoning-focused synthetic data addition enhances performance.** Our ReasonAQA experiments show that reasoning-focused training improves performance on reasoning tasks. When generating QA pairs for ReasonAQA, we rely on a generalized prompt for LLMs. However, one can also incorporate expert-written questions in the prompt to elicit more specific audio- and signal-centric reasoning queries. We observe that expert-prompt-generated questions (Type-2 and Type-3) are valuable, and combining them with ReasonAQA (Type-1) leads to further performance gains (Type-4). While additional QA pairs generated via expert-question prompts can continue to boost performance.

## 6 Conclusion

We introduce Mellow, a small audio-language model for reasoning, demonstrating that sub-billion parameter models can achieve state-of-the-art performance. Mellow is evaluated on diverse tasks, including multimodal audio understanding and reasoning, deductive reasoning, comparative reasoning, and audio captioning, and beats several larger models on the tasks. Through ablation studies, we identify key factors that improve reasoning in small models, including language model pretraining, projection layers, reasoning-focused synthetic data, and more. We hope this work inspires further research on improving reasoning in small audio-language models independent of data (audio) scaling.

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

# Appendix

## Table of Contents

# A   Responsible AI

## A.1   Ethics statement

The development and evaluation of Mellow prioritize fairness, transparency, and societal benefit. We acknowledge that models, especially those trained on multimodal data, can inherit biases present in their training datasets. To mitigate potential biases, we carefully curated the ReasonAQA dataset and removed any harmful generations from the data generation process. Mellow's primary applications are designed to enhance accessibility, improve audio understanding in real-world scenarios, and contribute to research in efficient, on-device AI models. However, we recognize that models of this nature could be misused for tasks such as unauthorized surveillance or generating misleading audio-text outputs. To address this, we release Mellow with explicit usage guidelines and encourage responsible AI practices. To support transparency, we openly share our dataset and benchmarks for external audits. Additionally, Mellow's lightweight design and efficient training methods help reduce the environmental impact of AI research.

## A.2   Reproducibility statement

We have implemented and thoroughly documented multiple measures to support the reproducibility of our work across the main paper, appendix, and supplementary materials. To this end, we make the following artifacts publicly available. (1) Model Checkpoints: We release the trained weights of Mellow to enable further research, fine-tuning, and benchmarking (2) Training Data: We provide the ReasonAQA dataset, including the synthetic question-answer pairs derived from AudioCaps and Clotho, along with details on how the dataset was generated using large language models. (3) Experimental Details: We document all key hyperparameters, model architectures, and training procedures, allowing researchers to replicate our training pipeline. (4) Ablation Studies: We present thorough ablation studies on projection layers, language model pretraining, and synthetic data generation to provide insights into the model's design choices (5) All experiments were conducted on well-documented benchmarks, ensuring comparability with existing methods. Furthermore, to promote transparency, we include error analyses highlighting areas where Mellow underperforms, such as speech-related reasoning tasks, and provide insights into potential improvements. By sharing these resources, we aim to enable the broader research community to build upon our results, findings, and further develop efficient audio-language reasoning models.

# B   Related work

**Audio-text Learning.** Recently, text has been increasingly used as a supervisory signal for learning audio representations. This has led to the development of two primary learning approaches: contrastive audio-text learning [25] and audio-conditioned next-token prediction [14]. Contrastive methods produce audio-language models capable of zero-shot classification and retrieval at test time, while audio-conditioned next-token prediction enables models to perform open-ended tasks such as audio question answering. In parallel, task-specific models that leverage language are also being developed, including models designed for audio captioning [23, 53], audio-text retrieval [15], and text-to-audio generation [49].

**Audio-Language Models.** With these two pretraining methods, the field is moving toward general-purpose Audio-Language Models. These models are pretrained on millions of audio-text pairs and can be prompted at test time to perform multiple tasks. Contrastive Audio-Language Models [25, 27, 73] achieve state-of-the-art (SoTA) performance on closed-ended audio tasks such as classification and retrieval, surpassing task-specific models. Similarly, generative Audio-Language Models [14, 36, 33, 34, 10, 11, 66] achieve SoTA performance on open-ended tasks such as audio captioning and audio question answering. Over the years, research has focused on improving audio encoders [66, 33], language models [27], and pretraining and post-training strategies [10, 11, 19]. A consistent trend across these improvements has been increasing both the training data and language model parameters, enabling ALMs to acquire novel capabilities previously unseen in smaller-scale models.

**Audio reasoning.** Scaling Audio-Language Models in terms of both data and compute has led to the emergence of novel abilities [16, 18] that were not explicitly trained for. In real-world scenarios, these models must process diverse types of queries, requiring them to listen (perceive) to the audio, understand the user's question, integrate world knowledge, and reason over both audio-text information and external knowledge to formulate responses. Enhancing the reasoning capabilities of

Audio-Language Models can significantly improve performance across multiple tasks. Consequently, recent studies [18, 19, 63] have focused on benchmarking reasoning ability. These benchmarks evaluate various reasoning skills [63] and different types of logical reasoning [18, 19], such as deductive reasoning, inductive reasoning, and comparative reasoning.

**Small Language Models.** In recent years, there has been a growing emphasis on developing small language models that maintain strong performance while significantly reducing computational overhead. This research direction has given rise to a range of models, including the Phi series [37, 47, 40, 1], the smolLM series [2], OpenELM [52], MobileLLM [51], and others. These models employ techniques such as knowledge distillation, embedding and block-wise weight sharing, training on curated synthetic and textbook data, deep yet narrow architectures, grouped-query attention mechanisms, and quantization. These approaches enable small models to achieve performance comparable to larger models while maintaining lower memory requirements and reduced energy consumption, making real-time, on-device inference feasible.

**Small Audio-Language Models.** The prevailing trend in audio-language modeling has been to scale up training data and computational resources (i.e., language model parameters), leading to unified models [66, 10, 11] capable of understanding and reasoning across diverse audio domains, including music and speech. However, it is essential to explore methods for enhancing comprehension and reasoning while operating under limited data and computational constraints. Models with fewer than 1 billion language model parameters, referred to as Small Audio-Language Models, offer unique advantages in terms of memory and energy efficiency, making on-device inference possible. Currently, the only Small Audio-Language Model in the literature, Pengi [14], exhibits known limitations in performing open-ended tasks such as audio question answering [36] and reasoning-based tasks [63, 18, 19]. In this paper we explore and push the ceiling of Small Audio-Language Model performance.

# C   ReasonAQA

The construction of ReasonAQA is described in Section 2. The data construction process consists of three main components. First, we select audio sources to build ReasonAQA around. This includes audio files from AudioCaps and Clotho, ensuring the dataset covers a wide range of audio events, acoustic scenes, and audio concepts. Additionally, by restricting the dataset to these two audio sources, we maintain consistency in audio concepts and isolate performance improvements from data scaling. Second, we incorporate existing audio reasoning datasets from the literature, such as audio entailment and audio difference explanation, and convert them into the AQA format for training. Third, we generate a synthetic dataset for the selected audio sources in ReasonAQA. This synthetic data constitutes 70% of the total training dataset for ReasonAQA.

## C.1   Generating synthetic data

In this section, we describe the third part of our data construction process: generating a synthetic training dataset for ReasonAQA. The process is illustrated in Figure 4. Our goal with synthetic data generation is to create questions that are grounded in audio, elicit reasoning, and include a variety of question types, resulting in both detailed and multiple-choice (MCQ) answers. We use Llama 3 8B [24], an open-source model with static weights, which enables the ReasonAQA pipeline to be reproduced on consumer-grade GPUs. The resulting synthetic dataset constitutes approximately 70% of the total OpenAQA training data, with 35% allocated to detailed QA (333k instances) and 35% to MCQ QA (337k instances).

**Method.** We use audio descriptions from AudioCaps and Clotho to prompt an LLM to generate question-answer (QA) pairs. The process is illustrated in Figure 4. This approach of LLM-based prompting has been widely used to create various datasets and benchmarks [57, 36, 18, 19].

**Detail AQA.** The detailed AQA subset consists of question-answer pairs where the questions require detailed responses that incorporate audio events, acoustic scenes, signal characteristics, and their compositional and temporal relationships. To achieve this, we prompt the LLM with example questions such as: "what is the sound event present in the clip?", "what should I do when I hear this sound?", "what is this sound most similar to?". This encourages the LLM to generate diverse question-answer pairs covering audio events, acoustic properties, psychological impact, and signal characteristics. The prompt used for generating this data is shown in Figure 5.

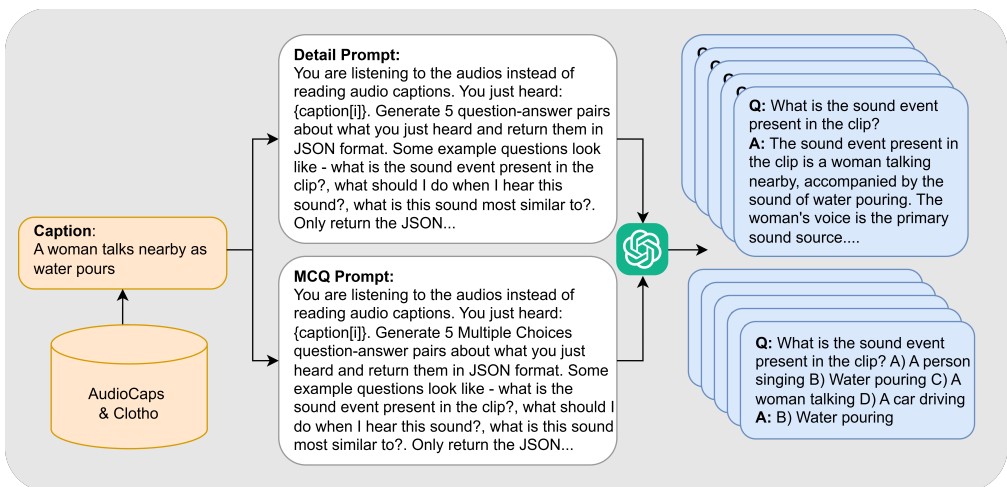

Figure 4: The data generation pipeline for creating the training data of ReasonAQA consists of three main steps. First, audio captions are sampled from the AudioCaps [43] and Clotho [23] datasets. Next, these audio captions are inserted into detailed and multiple-choice (MCQ) templates to construct text prompts. Finally, these text prompts are used to query a large language model (LLM), which generates detailed and MCQ-based audio question-answer pairs.

**MCQ AQA.** The multiple-choice AQA subset consists of question-answer pairs where each question includes multiple-choice options, and the answer is a single selected option. MCQ-based questions are easier to evaluate due to their classification nature, avoiding the challenges associated with evaluating open-ended responses. Furthermore, by explicitly providing answer options within the question, we mitigate issues related to ambiguous or correct-yet-irrelevant answers. The MCQ questions cover a wide range of audio concepts, including audio events, acoustic scenes, signal characteristics, and their compositional and temporal relationships. The prompt structure is similar to that of the detailed QA generation process, with example questions such as: "What sound event is present in the clip?", "What should I do when I hear this sound?", "What is this sound most similar to?". The exact prompt used for MCQ generation is shown in Figure 5.

```
{
    "system_prompt": "You are a helpful assistant with expert knowledge about audio, acoustics, and
    ↪ psychoacoustics. You study audio, which is the study of sound and its properties. You study
    ↪ acoustics, which revolve around the generation, propagation, and reception of sound waves. You
    ↪ study Psychology which posits that a sound is a complex stimulus that encompasses a vast range
    ↪ of acoustic properties involving aspects of cognition, psychoacoustics, and psychomechanics.
    ↪ Your task is to generation question-answer pairs based on the captions of audio content using
    ↪ natural language. To answer the questions, you utilize words related to their acoustic
    ↪ properties, such as their semantic relations, their spectro-temporal characteristics,
    ↪ frequency, loudness, duration, materials, interactions, and sound sources",

    "user_prompt_detail": "You are listening to the audios instead of reading audio captions. You just
    ↪ heard: {caption[i]}. Generate 5 question-answer pairs about what you just heard and return
    ↪ them in JSON format. Provide answers in detail. Some example questions to take inspiration
    ↪ from - what is the sound event present in the clip?, what should I do when I hear this sound?,
    ↪ what is this sound most similar to?. Only return the JSON where each dictionary contains a
    ↪ question key and an answer key",

    "user_prompt_mcq": "You are listening to the audios instead of reading audio captions. You just
    ↪ heard: {caption[i]}. Generate 5 Multiple Choices question-answer pairs about what you just
    ↪ heard and return them in JSON format. Provide answers in detail. Some example questions look
    ↪ like - what is the sound event present in the clip?, what should I do when I hear this sound?,
    ↪ what is this sound most similar to?. Only return the JSON where each dictionary contains
    ↪ question key, choices key with alphabets and an answer key with alphabets."
}
```

Figure 5: LLM system prompt used to generate MCQ and descriptive questions for ReasonAQA. The "user prompt detail" and "user prompt mcq" shows the prompt used to generate the detailed and MCQ audio question-answer pairs respectively. In Ablation Table 14, this is referred to as Type 1 data generation.

## C.2   Analysis of synthetic data

In this section, we analyze the data generated by the process described in C.1, which constitutes the synthetic portion of ReasonAQA. Specifically, this process uses an LLM to create multiple-choice (MCQ) and detailed question-answer pairs based on audio captions sourced from the AudioCaps and Clotho datasets.

**Data distribution**. We first examine the data distribution for the MCQ and detailed synthetic data, with Table 9 reporting the word length and vocabulary size for both questions and answers. The ReasonAQA dataset comprises 209,412 multiple-choice (MCQ) and 222,499 detailed question-answer pairs, sourced from AudioCaps and Clotho. MCQs have an average question length of 11.39 words with a vocabulary size of 10,035, while their answers are shorter, averaging 7,356 words with a limited vocabulary of 2.31 words. In contrast, detailed questions are significantly shorter (3.01 words on average) but exhibit much greater vocabulary diversity (25.65 words), with their answers being substantially longer, averaging 16,161 words. In all, the MCQs prioritize concise, factual understanding, whereas detailed QA pairs emphasize linguistic richness and deeper reasoning, making ReasonAQA well-suited for training models on both structured knowledge retrieval and complex audio reasoning.

| Category | Dataset | # of QA pairs | Question | | Answer | |
| --- | --- | --- | --- | --- | --- | --- |
| | | | Length (# of words) | Vocab. (# of words) | Length (# of words) | Vocab. (# of words) |
| MCQ | AC | 113861 | 11.33 | 7697 | 2.28 | 5411 |
| | Clotho | 95551 | 11.46 | 8430 | 2.35 | 6009 |
| | Overall | 209412 | 11.39 | 10035 | 2.31 | 7356 |
| Detail | AC | 127661 | 3.06 | 3034 | 25.90 | 13181 |
| | Clotho | 94838 | 2.95 | 3443 | 25.32 | 13445 |
| | Overall | 222499 | 3.01 | 4295 | 25.65 | 16161 |

Table 9: Data statistics of synthetically generated data for ReasonAQA

**Data vocabulary.** Since the data is generated by an LLM, it is essential to analyze the characteristics of the generated QA pairs. By tokenizing the text and identifying audio-related words, we determine the most frequently occurring terms in both MCQ and detailed question-answer pairs. The results of this analysis are presented in Fig. 6, highlighting three primary areas of focus:

- **Acoustic properties.** Both questions and answers frequently include terms related to fundamental acoustic characteristics. Words such as *frequency, range, loud, loudness, pitch, Hz*, and descriptive attributes like *low, mid, high, gentle, soft* emphasize the analysis of sound's physical properties. Questions explicitly reference these aspects using terms like *acoustic, psychoacoustic, frequency, loudness, pitched*, while answers reflect similar vocabulary.

- **Sound events and composition.** Another major focus of the dataset is the identification and description of sound events within the audio. Questions often include terms like *event, present, primary, background, sounds, noise*, prompting the model to describe the composition of the soundscape. Similarly, answers feature words such as *primary, background, present, noise*, highlighting how different elements within the audio contribute to the overall scene.

- **Inference and perceptual understanding.** The dataset also contains questions and answers that require higher-level reasoning beyond direct acoustic analysis. For example, answers contain Words like *likely, sense, similar, may*, which suggests responses involve making inferences or drawing conclusions based on the audio. Likewise, questions containing terms such as *emotional, similar, likely, sense* are geared towards inferring meaning, context, and subjective perception from audio.

A randomly sampled detailed and MCQ example from ReasonAQA is shown in Table 10 and Table 11, respectively.

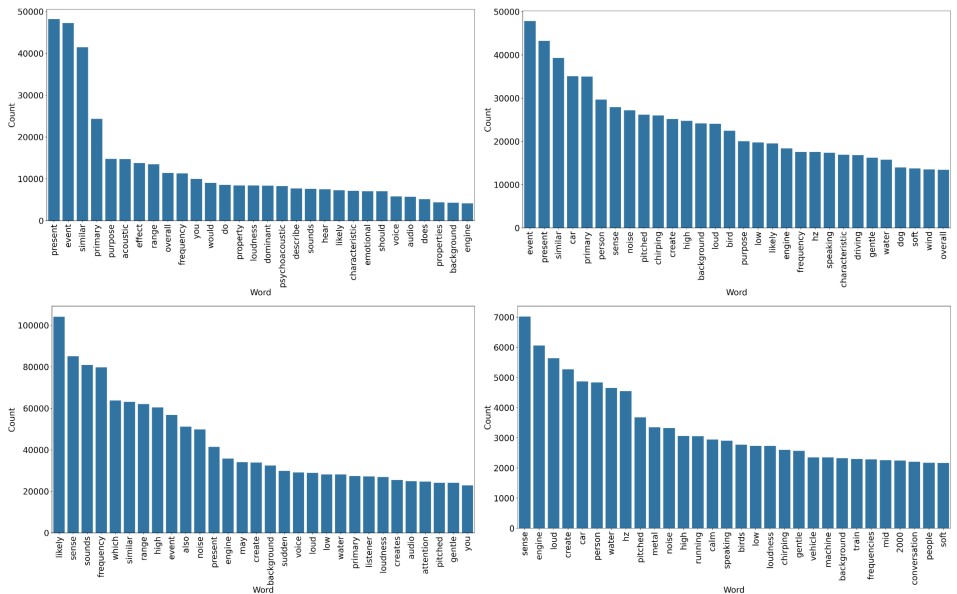

Figure 6: Comparison of most frequent words in detail questions (top left), detail answers (top right), MCQ questions (bottom left) and MCQ answers (bottom right)

---

**Caption: The wind is blowing and rustling occurs**

**Question:** What is the sound event present in the clip?
**Answer:** The sound event present in the clip is the rustling of wind through leaves or other materials, likely creating a gentle, soothing background noise.
**Question:** What is the primary characteristic of this sound?
**Answer:** The primary characteristic of this sound is its gentle, soft quality, with a subtle variation in intensity and pitch due to the changing direction and speed of the wind.
**Question:** What should I do when I hear this sound?
**Answer:** When you hear this sound, you might want to take a moment to relax and calm down, as the gentle rustling of the wind can have a soothing effect on the mind and body.
**Question:** What is this sound most similar to?
**Answer:** This sound is most similar to the gentle rustling of leaves or grass in a breezy day, or the soft whispering of a gentle stream or river.
**Question:** What acoustic properties are present in this sound?
**Answer:** This sound exhibits properties of gentle amplitude modulation, with subtle changes in frequency and amplitude due to the varying wind speed and direction, creating a sense of movement and texture.

Table 10: ReasonAQA detail type examples for sampled audio events.

---

**Caption: The wind is blowing and rustling occurs**

**Question:** What is the primary sound event present in the clip? A) Bird chirping B) Wind blowing C) Rain falling D) Ocean waves crashing
**Answer:** B) Wind blowing
**Question:** What is the characteristic of the sound in the clip? A) Sharp and piercing B) Soft and gentle C) Rustling and whooshing D) Crashing and booming
**Answer:** C) Rustling and whooshing
**Question:** What is the sound most similar to? A) A fan blowing B) A leaf blower C) The wind blowing through trees D) A vacuum cleaner
**Answer:** C) The wind blowing through trees
**Question:** What is the sound indicating? A) A storm is approaching B) The weather is calm C) A strong gust of wind is coming D) A tornado is forming
**Answer:** B) The weather is calm
**Question:** What should you do when you hear this sound? A) Take cover and seek shelter B) Go outside and enjoy the weather C) Check the weather forecast D) Close all windows and doors
**Answer:** B) Go outside and enjoy the weather

Table 11: ReasonAQA MCQ type examples for sampled audio events.

# D   Comparing with OpenAQA

The most widely used audio question-answering dataset in the literature is OpenAQA [35]. OpenAQA was created using GPT-3.5 Turbo to generate audio-based question-answer pairs from audio metadata. It comprises 845K unique audio clips sourced from datasets such as AudioSet [31], AudioSet-SL, VGGSound [8], FreeSound [30, 29], AudioCaps [43], Clotho [23], and SoundBible [15]. In total, OpenAQA contains 5.6 million QA pairs, including 3.7 million open-ended questions and 1.9 million close-ended questions. It is widely used for training audio-language models such as LTU [35], LTU-AS [34], GAMA [33], and many more.

---

**Answerable by World Knowledge**

---

**Sample:** The sizzling of food while a dish is clanking
**Question:** What kind of sound is sizzling?
**Answer:** Sizzling is the high-pitched sound that results from food being cooked on a hot surface.
**Question:** What is the significance of the sizzling sound in cooking?
**Answer:** The sizzling sound in cooking indicates that food is being cooked properly and that the temperature of the cooking surface is sufficiently high.

---

**Refused Answers**

---

**Sample:** Person is whistling
**Question:** What is the duration of the whistling?
**Answer:** The duration of the whistling is unknown.
**Question:** What is the pitch of the whistle?
**Answer:** It is unclear what the pitch of the whistle is.

---

**Repetitive Questions**

---

**Sample:** The sizzling of food while a dish is clanking
**Question:** Does the audio clip convey a sense of urgency or emergency?
**Answer:** No, the audio clip does not convey any sense of urgency or emergency, as it is simply the sounds of food being cooked.
**Question:** What time of day could this audio clip be recorded?
**Answer:** The audio clip could be recorded during any time of the day or night, as cooking is a common activity that can occur at any time.

---

**Bad Generations**

---

**Sample:** Someone has a hiccup while typing
**Question:** How might the sound events in the audio clip affect someone who hears it?
**Answer:** It's difficult to determine how someone else might react to this audio clip, as it depends on the individual's perception and interpretation of the sounds.
**Question:** What might the perception of the audio clip change if other sounds were added to it?
**Answer:** The addition of other sounds would likely affect how the audio clip is perceived and interpreted, but it's difficult to say what those sounds might be without more information.

---

Table 12: Categories of audio question generation issues in OpenAQA

## D.1   Limitations of OpenAQA

OpenAQA is the largest open-source audio question-answering dataset and is widely used in the audio literature to train audio-language models. However, a closer examination reveals several issues with the generated question-answer pairs, as illustrated in Table 12. These issues can be categorized into four types:

- **Answerable by World Knowledge.** A significant portion of the generated questions do not require any audio-specific reasoning and can be answered solely using world knowledge. For instance, the question "What kind of sound is sizzling?" results in an answer that merely defines the term "sizzling" rather than analyzing the actual content of the audio. Such data does not ellict audio-grounded reasoning in the model.

- **Refused Answers.** The generated questions frequently receive refusal as the answer, even when the necessary information is present in the audio. For example, in response to questions about the sound of a person whistling, the reference answers include "The duration of the whistling is unknown." and "It is unclear what the pitch of the whistle is." Despite these aspects being extractable from the audio, the reference answer is a refusal. Such responses prevent the model from learning any acoustic features of the audio. While refusals are appropriate when a question cannot be answered based on the audio alone, OpenAQA frequently refuses to answer even when a valid response is possible. As a result, large portions of the dataset contain refusals rather than meaningful acoustic reasoning.

- **Repetitive Questions.** OpenAQA contains numerous redundant questions across different audio clips. Questions such as "Does the audio clip convey a sense of urgency or emer-

gency?" and "What time of day could this audio clip be recorded?" appear frequently, even when they are not meaningful or contextually relevant to the given audio.

- **Bad Generations.** OpenAQA also suffers from poorly formulated or irrelevant questions. For example, a question like "What might the perception of the audio clip change if other sounds were added to it?" is vague and does not meaningfully engage with the specifics of the provided audio. This issue likely arises because the LLM is required to generate ten questions per caption, leading to an increase in illogical or poorly structured questions.

---

**Caption: Someone fires a gun with others talking in the background**

---

*OpenAQA examples*

**Question:** What is the acoustic feature that allows you to differentiate between the gunshot and gunfire?
**Answer:** Gunshots and gunfire differ in their duration and intensity. Gunshots have a shorter duration and are louder compared to gunfire. Therefore, the acoustic feature that differentiates the two is their duration and intensity.
**Question:** What is the potential scenario that this audio clip represents, given the combination of sound events?
**Answer:** The scenario could be a shooting, where someone fires a gun and others in the background react by talking.
**Question:** What is the temporal relationship between the gunshot/gunfire and the speech in the audio clip?
**Answer:** The gunshot/gunfire happens before the speech in the audio clip. This is because the person fires the gun first, and then others in the background react by talking.
**Question:** What can be inferred about the mood or atmosphere that this audio clip conveys?
**Answer:** The mood or atmosphere could be tense and chaotic since there is a gunshot/gunfire, and people are talking in the background.
**Question:** What can you conclude about the proximity of the person firing the gun based on the audio clip?
**Answer:** It is difficult to determine the proximity of the person firing the gun based on the audio clip alone since there is no information regarding microphone placement and other environmental factors.
**Question:** How can you tell whether the gun was fired indoors or outdoors based on the audio clip?
**Answer:** It is difficult to tell whether the gun was fired indoors or outdoors based solely on the audio clip since there is no information about ambient noise or echoes.
**Question:** Who do you think fired the gun based on the audio clip alone?
**Answer:** It is impossible to determine who fired the gun based solely on the audio clip since there is no information regarding the person, their location, or the circumstances leading up to the gunshot.
**Question:** What is the difference between gunfire and a single gunshot based on their acoustic features?
**Answer:** Gunfire refers to multiple gunshots. Gunfire tends to have a longer duration and a lower intensity compared to a single gunshot, which is louder and has a shorter duration.
**Question:** What can you infer about the location of the other people in the audio clip who are talking in the background?
**Answer:** It is impossible to determine the location of the other people in the audio clip based solely on the audio clip since there is no information regarding their location or the environment.
**Question:** What could be the reason for firing a gun in the scenario represented by the audio clip?
**Answer:** It is impossible to determine the reason for firing a gun in the scenario represented by the audio clip based solely on the audio clip since there is no context or additional information provided.

---

*ReasonAQA examples*

**Question:** What is the sound event present in the clip?
**Answer:** The sound event present in the clip is a gunshot, accompanied by background conversations and ambient noise. The gunshot is a sudden, sharp, and loud sound, followed by a decaying reverberation, indicating a sudden release of energy. The background conversations and ambient noise provide a sense of context, suggesting a chaotic or noisy environment.
**Question:** What is the dominant acoustic property of the gunshot?
**Answer:** The dominant acoustic property of the gunshot is its high-frequency content, particularly in the range of 5 kHz to 20 kHz. This is because the sudden release of energy from the gunpowder creates a high-pressure wave that propagates through the air, producing a sharp, crisp sound with a high-frequency emphasis.
**Question:** How does the gunshot sound different from the background conversations?
**Answer:** The gunshot sounds different from the background conversations in terms of its spectral characteristics, duration, and intensity. The gunshot has a much higher intensity, with a peak amplitude that is significantly higher than the background conversations. The gunshot also has a much shorter duration, lasting around 0.1-0.2 seconds, whereas the background conversations are more sustained and have a longer duration.
**Question:** What is the psychological impact of hearing this sound?
**Answer:** Hearing this sound can have a significant psychological impact, particularly due to its association with violence, danger, or sudden, intense events. The sudden and intense nature of the gunshot can trigger a fight-or-flight response, releasing stress hormones such as adrenaline and cortisol. The background conversations may also contribute to a sense of chaos or confusion, making it difficult to process the information and respond appropriately.
**Question:** What is this sound most similar to?
**Answer:** This sound is most similar to other sudden, intense sounds, such as firecrackers, car crashes, or explosions. The sharp, high-frequency content and sudden release of energy are common characteristics of these types of sounds, making them easily recognizable and evocative of strong emotions and reactions.

---

Table 13: Comparing OpenAQA examples with ReasonAQA examples. From both dataset, we sample detail questions per audio event.

## D.2 Qualitative comparison

Both OpenAQA and ReasonAQA contain open-ended QA data, making OpenAQA somewhat comparable to the detailed question-answer pairs in ReasonAQA. Table 13 presents a direct comparison

of the generated QA pairs for the same audio file in both datasets. One key distinction is that ReasonAQA provides longer, more detailed answers for open-ended questions, allowing the model to generate richer and more informative responses. Additionally, ReasonAQA reduces the number of generated questions per iteration from ten to five, prioritizing quality over quantity and mitigating the redundancy and poorly formulated questions observed in OpenAQA. Unlike OpenAQA, ReasonAQA also includes multiple-choice (MCQ) question-answer pairs. By presenting multiple answer choices, the model learns to focus on selecting the correct option, thereby improving its reasoning ability. Furthermore, ReasonAQA offers more precise descriptions of acoustic properties such as frequency ranges, loudness, and duration, making it a more structured and informative dataset for training audio-language models.

# E   Experimental setup

## E.1   Architecture

**Audio encoder.** The audio sampling rate is 32 kHz and we use HTSAT [9] as the audio encoder. HTSAT truncates the audio to 10 seconds and produces three outputs: framewise, clipwise, and latent. The framewise output ($t \times c$) are the time-presence probabilities for AudioSet classes and the clipwise output ($1 \times c$) are per-class probabilities averaged across time. The latent output ($1 \times l$) is the hidden state output before expanding to framewise probabilities using token-semantic CNN.

**Mapper.** The output of the audio encoder is projected into the language model space using a mapper (middle subplot in Figure 2). For Mellow, we use the framewise output ($t \times c$) to retain temporal information and the latent output ($1 \times l$) as the audio summary and equivalent of a CLS token [20]. The framewise output is projected using a linear layer and concatenated with latent output, leading to the audio embedding output ($t + 1 \times l$). This audio embedding output is passed to the projection layer. The output of the projection layer is then $8\times$ downsampled using 2D average pooling. While downsampling, we retain the latent (CLS) output.

**Projection.** The projection layer consists of two linear layers, whose outputs are merged followed by LayerNorm. This is shown on the right side of Figure 2.

## E.2   Training

Mellow is trained using next-token prediction where the model predicts the next-text token conditioned on audio and text input. The input to the model is audio 1 ($x_1^i$), audio 2 ($x_2^i$), text prompt ($t^i$) and the output is text ($c^i$). The audios are encoded by an audio encoder ($a_\phi$) and the text prompt is embedded by text embedder ($g_\psi$). The audio embeddings are projected using a mapping network ($m_\zeta$), and then concatenated with the input text embedding. During concatenation, we add a separator token ($s$) between audio 1, audio 2, and text embedding.

$$p^i = p_1^i, ..., p_k^i = \text{concat}\{m_\zeta(a_\phi(x_1^i)), s, m_\zeta(a_\phi(x_2^i)), s, g_\psi(t^i)\} \tag{2}$$

The total prefix $\{p_j^i\}_{j=1}^k$ is then used to prompt the language model ($f_\theta$) to produce text output. The model learns to predict next-token $o^i$ based on the prefix $p^i$. The loss is summation of cross-entropy over tokens:

$$\mathcal{L} = -\sum_{i=1}^{N} \sum_{j=1}^{l} \log p_\gamma(o_j^i | p_1^i, ..., p_k^i, o_1^i, ..., o_{j-1}^i) \tag{3}$$

where $\gamma$ are Mellow's trainable parameters and consists of $\zeta, \theta$. We use Adam Optimiser [44] with cosine learning rate schedule with a maximum learning rate of 1e-3. We train Mellow and all the ablation models for 30 epochs.

## E.3   Inference

We use top-p sampling where p is set to 0.8 and a temperature of 1.

## E.4   Evaluation

We use different evaluation metrics depending on the task, categorizing them into close-ended and open-ended tasks.

**Close-ended tasks.** For close-ended tasks such as classification and multiple-choice questions (MCQ), we use standard evaluation metrics, including Accuracy, Precision, Recall, and F1-score. These metrics are applied in tasks like Audio Entailment [18], ClothoAQA [48], and MMAU [63].

**Open-ended tasks.** For descriptive tasks, such as audio difference explanation [19] and audio captioning [43, 23], we use text-based evaluation metrics, including BLEU [59], METEOR [4], SPICE [3], CIDEr [71], and SPIDEr [50], to compare generated responses against ground-truth descriptions. BLEU (Bilingual Evaluation Understudy) assesses the precision of n-grams between generated and reference text, providing a simple and efficient approach, though it does not account for recall, word order, or deeper semantic meaning. SPICE (Semantic Propositional Image Caption Evaluation) improves upon BLEU by analyzing the semantic content of text using scene graphs, capturing deeper meaning rather than relying on surface-level n-gram matches, though at a higher computational cost. SPIDEr, a combination of SPICE and CIDEr, integrates both semantic and consensus-based evaluation, balancing precision with deeper content understanding. Since the existing Audio-Language Model (ALM) literature predominantly uses SPICE, we adopt the same metric to ensure a fair comparison with prior work.

# F  Ablation studies

We perform ablation studies to identify which components most effectively enhance reasoning in audio-language models, with a particular focus on Small ALMs. In particular, we examine the effects of audio encoders (Section F.2), language model choice (Section F.3), projection layers (Section F.1), prefix-tuning versus fine-tuning (Section F.4), LoRA adaptation (Section F.5), synthetic data generation (Section F.6), and scaling audio data (Section F.7). The results are shown in Table 8.

| Models | Size | Audio Caption | | B-AQA | Audio Entailment | | Audio Difference | | MMAU (test-mini) | | | |
| | | AC (SPICE) | CL (SPICE) | ClothoAQA (ACC) | CLE (ACC) | ACE (ACC) | CLD-3 (SPICE) | ACD-3 (SPICE) | Sound (ACC) | Music (ACC) | Speech (ACC) | Avg. (ACC) |
|---|---|---|---|---|---|---|---|---|---|---|---|---|
| *The SLM is GPT2 and frozen and the projection layer is changed* | | | | | | | | | | | | |
| Linear | 156M | 4.53 | 5.98 | 53.23 | 30.79 | 30.15 | 5.21 | 8.39 | 16.80 | 31.25 | 30.10 | 26.05 |
| Non-linear 2 | 157M | 4.79 | 6.10 | 55.18 | 32.12 | 29.56 | 5.56 | 8.62 | 17.42 | 33.23 | 27.03 | 25.89 |
| Transformer [14] | 195M | 10.26 | 7.89 | 62.35 | 43.89 | 42.10 | 11.88 | 12.50 | 28.65 | 35.23 | 26.68 | 30.19 |
| *The SLM is GPT2 and finetuned and the projection layer is changed* | | | | | | | | | | | | |
| Linear | 156M | 9.87 | 7.10 | 70.90 | 93.45 | 92.90 | 13.65 | 14.21 | 47.64 | 45.39 | 27.00 | 40.01 |
| Non-linear 2 | 157M | 10.51 | 7.15 | 71.25 | 93.40 | 93.27 | 13.77 | 14.01 | 48.05 | 48.50 | 27.33 | 41.29 |
| Transformer [14] | 195M | 10.77 | 7.23 | 70.89 | 92.32 | 93.65 | 13.45 | 15.21 | 47.89 | 49.10 | 27.10 | 41.36 |
| *The SLM is frozen, projection layer is non-linear and the SLM is changed* | | | | | | | | | | | | |
| GPT2 frozen | 157M | 4.79 | 6.10 | 55.18 | 32.12 | 29.56 | 5.56 | 8.62 | 17.42 | 33.23 | 27.03 | 25.89 |
| SmolLM2 frozen | 167M | 5.26 | 8.58 | 45.70 | 33.40 | 31.66 | 11.43 | 13.62 | 35.14 | 30.54 | 20.12 | 28.60 |
| *The SLM is finetuned, projection layer is non-linear and the SLM is changed* | | | | | | | | | | | | |
| GPT2 finetune | 157M | 10.51 | 7.15 | 71.25 | 93.40 | 93.27 | 13.77 | 14.01 | 48.05 | 48.50 | 27.33 | 41.29 |
| SmolLM2 finetune | 167M | 18.60 | 9.83 | 71.65 | 92.00 | 90.85 | 17.33 | 18.68 | 59.46 | 50.60 | 28.82 | 46.29 |
| *Diferent finetuning methods. The SLM is SmolLM2, projection layer is non-linear* | | | | | | | | | | | | |
| Prefix-tuning | 167M | 5.26 | 8.58 | 45.70 | 33.40 | 31.66 | 11.43 | 13.62 | 35.14 | 30.54 | 20.12 | 28.60 |
| LoRA (8, 16) | 167M | 18.53 | 9.25 | 64.64 | 79.33 | 84.15 | 14.23 | 14.98 | 45.95 | 42.51 | 28.83 | 39.10 |
| LoRA (256, 512) | 181M | 19.01 | 10.59 | 65.54 | 86.66 | 89.87 | 15.36 | 16.51 | 50.75 | 49.10 | 33.33 | 44.40 |
| Finetuning | 167M | 18.60 | 9.83 | 71.65 | 92.00 | 90.85 | 17.33 | 18.68 | 59.46 | 50.60 | 28.82 | 46.29 |
| *The SLM is SmolLM2 and finetuned, projection layer is non-linear and the audio encoder is changed* | | | | | | | | | | | | |
| CNN14 | 219M | 15.97 | 7.91 | 65.82 | 91.06 | 92.39 | 16.27 | 16.95 | 54.05 | 47.60 | 28.23 | 43.30 |
| HTSAT | 167M | 18.60 | 9.83 | 71.65 | 92.00 | 90.85 | 17.33 | 18.68 | 59.46 | 50.60 | 28.82 | 46.29 |
| *The SLM is SmolLM2 and finetuned, projection layer is non-linear, and training data is changed* | | | | | | | | | | | | |
| Type 1 | 167M | 18.60 | 9.83 | 71.65 | 92.00 | 90.85 | 17.33 | 18.68 | 59.46 | 50.60 | 28.82 | 46.29 |
| Type 2 | 167M | 16.47 | 8.23 | 71.05 | 92.50 | 93.20 | 16.98 | 18.09 | 59.45 | 42.81 | 37.84 | 46.70 |
| Type 3 | 167M | 17.43 | 9.88 | 66.83 | 91.87 | 94.25 | 17.37 | 18.67 | 61.56 | 45.21 | 32.43 | 46.40 |
| Type 4 | 167M | 17.79 | 9.38 | 71.39 | 91.16 | 89.66 | 17.21 | 18.54 | 61.26 | 54.19 | 29.73 | 48.40 |
| *The SLM is SmolLM2 and finetuned, projection layer is non-linear, and WavCaps is added to training* | | | | | | | | | | | | |
| Type 1 | 167M | 18.60 | 9.83 | 71.65 | 92.00 | 90.85 | 17.33 | 18.68 | 59.46 | 50.60 | 28.82 | 46.29 |
| + WavCaps | 167M | 14.83 | 9.66 | 71.32 | 92.47 | 92.69 | 17.92 | 19.13 | 59.16 | 60.48 | 23.72 | 47.80 |
| **Mellow** | 167M | 17.79 | 9.38 | 71.39 | 91.16 | 89.66 | 17.21 | 18.54 | 61.26 | 54.19 | 29.73 | 48.40 |

Table 14: Ablation study results. For the reader's convenience, we reproduce here the same table presented as Table 8 .

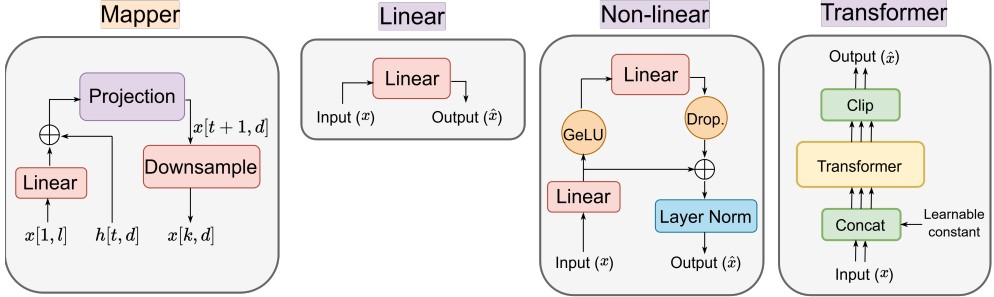

Figure 7: The common type of projection layers used in literature. The Mapper used in Mellow is shown on the left side. For the projection layer in the mapper, we ablate with different popular projection layer structures – simple linear, non-linear, and transformer with learnable constant.

## F.1 Projection layer

We experiment with different projection layers to assess their impact on model performance. In previous studies involving frozen language models, the literature [14, 19, 17] has primarily used a stack of transformer layers with learnable constants. More recently, Q-Former [33] has been introduced, which increases the number of learnable constants while also making them query-dependent. In our experiments, we explore the use of linear, non-linear, and transformer-based projections in the mapper. For consistency across projection ablations, the linear layer and downsampling in the mapper remain fixed and are designed as a function of the chosen audio encoder.

The ablation results, presented in Table 14, indicate that transformer-based projection consistently outperforms both linear and non-linear projections when the language model is kept frozen. Since the projection layer is primarily responsible for steering the frozen language model toward generating the desired output, a higher parameter count in the projection layer becomes essential for effectively guiding Audio-Language Models (ALMs). However, when the language model is trained instead of frozen, the advantage of a high-parameter projection layer diminishes, and simpler linear or non-linear projection layers perform comparably.

## F.2 Choice of audio encoder

Audio encoders are trained on large-scale audio datasets using either self-supervised learning or cross-entropy loss over labeled data. In the audio literature, pretraining audio encoders on AudioSet in a supervised setup has been a common practice [14, 25, 27, 33, 36]. These pretrained audio encoders are trained to predict 527 audio events, and their latent representations can be effectively applied to various downstream tasks across sound and music using a linear-probe setup, where a linear layer is trained to classify downstream task labels. For Audio-Language Models (ALMs), we can assume that higher performance on AudioSet and improved linear-probe performance on downstream tasks indicate better separability in embeddings, which, in turn, enhances reasoning capabilities by covering a broader range of audio concepts.

To evaluate this, we conduct an ablation study using CNN14 [46] and HTSAT [9] as the chosen audio encoders. CNN14 is an 81M-parameter CNN-based model released in 2019, achieving 43.4 mAP on AudioSet and serving as one of the seminal models in audio literature. In contrast, HTSAT is a 31M-parameter transformer-based model, released in 2022, achieving 47.1 mAP on AudioSet and previously used as an audio encoder in ALM literature [25, 14]. Both models exhibit strong linear-probe performance, with HTSAT consistently outperforming CNN14. For the ablation study, we independently evaluate these audio encoders, followed by a mapper network and a non-linear projection, as illustrated in Figure 7. The linear layer dimensions in the mapper are set according to the latent dimensions of CNN14 and HTSAT. The ablation results, presented in Table 14, show that HTSAT outperforms CNN14 on MMAU, audio captioning, and binary AQA, while both models achieve comparable performance on audio difference and audio entailment tasks. These findings suggest that improving audio encoders enhances overall performance by increasing the coverage of audio concepts, thereby improving reasoning over a broader range of audio events. However, better audio encoders do not inherently enhance reasoning ability itself, as indicated by the similar performance of CNN14 and HTSAT on the audio entailment and audio difference tasks.

### F.3 Choice of SLM

In recent years, several Small Language Models (SLMs) have been developed [60, 2, 37, 47, 40, 1, 67, 68]. While the definition of what constitutes a "small" model has evolved over time—initially referring to models around 1B parameters and now extending up to 8B parameters—the performance of SLMs on reasoning tasks has shown continuous improvement. Our objective is to analyze whether the latest advancements in SLM pretraining and architectural modifications lead to improved reasoning performance for Audio-Language Models (ALMs).

In the audio-language modeling literature, the smallest existing Audio-Language Model is Pengi [14], which utilizes GPT-2, a ~125M parameter language model released in 2019. As a state-of-the-art (SoTA) SLM, we select SmolLM2 [2], which has a comparable parameter count (~135M). To isolate the effect of the language model, we keep the audio encoder, mapping network, and projection layer unchanged while replacing the language model. We evaluate both settings: keeping the language model frozen and fine-tuning it. The results, presented in Table 14, indicate that training with SmolLM2 outperforms GPT-2 in both scenarios, whether the language model is frozen or fine-tuned. Notably, in the audio difference explanation task—which requires the model to employ comparative reasoning to describe detailed (~155 words) differences between two audio samples—we observe a significant performance improvement when using a more advanced SLM.

### F.4 Freezing vs finetuning the LM

In this ablation study, we compare prefix tuning with fine-tuning the language model. Our results show that fine-tuning the language model consistently improves performance compared to keeping it frozen. While this outcome is expected, using a Small Language Model (SLM) makes full fine-tuning feasible, unlike Large Language Models (LLMs) with 8B+ parameters, where full fine-tuning is often impractical. Moreover, the performance improvement is substantial, as prefix-tuned models perform nearly at random on reasoning tasks.

If the language model must remain frozen, the mapping network needs to be proportionally increased in size relative to the language model to compensate for the lack of fine-tuning. Prior research [19] has demonstrated that a three-stage training process—unimodal pretraining, multimodal grounding, and fine-tuning—yields the best downstream performance. Previous studies relied on larger transformer-based projection layers, making the multimodal grounding stage essential. In contrast, we reduce the projection size by employing a simpler two-layer linear network and instead prioritize fine-tuning the model. This leads to a two-stage training process consisting of unimodal pretraining and fine-tuning. Our results indicate that using a non-linear mapper, we achieve performance comparable to—though slightly lower than—the three-stage training approach.

### F.5 LoRA adaptation vs finetuning the LM

In this ablation study, we compare LoRA adaptation [39] against full fine-tuning of the language model. LoRA has been employed by existing ALMs to fine-tune language models [36, 34, 33]. Following the ALM literature [36], we use a LoRA configuration with rank 8 and a scaling factor of 16, applied to the projection layers of the key and query in all self-attention blocks. For a 7B-parameter model, this setup introduces about 4.2M additional parameters; for a 135M-parameter model, it adds about 0.46M parameters.

Our results show that fully fine-tuning the language model consistently improves performance compared to LoRA adaptation. This is unsurprising, since full fine-tuning utilizes 135M trainable parameters, whereas LoRA adapts only 0.46M on the decoder side. Nevertheless, LoRA enables the model to retain broader world knowledge and the general capabilities of the ALM, thereby mitigating catastrophic forgetting. For example, the LoRA-adapted model can still answer questions such as "Who is the president of the United States?" or "What is the color of the sky?", tasks that may not be audio-related, which the full-finetuned model cannot. Retaining this general knowledge might be expected to improve the model's ability to handle new instructions and tasks; indeed, this appears partially true when the tasks are not strictly tied to audio. However, for audio-specific tasks, the LoRA-adapted model shows limited and comparable generalization to the fully fine-tuned LM.

While if one increases the rank and scaling factor, we see consistent improvement in performance. In cases like audio captioning performance, we see the LoRA model beating the full-finetuned model on SPICE metric. This can be attributed to retaining language model information and limitation

of SPICE metrics discussed in Table 18. The LoRA model also retain world-knowledge better as previously discussed, however, the improvements in LoRA come with increase in parameter count (167M to 181M). Therefore, the decision to use LoRa or not, depends on the goal to be achieved with the model. If the model uses a small LM and used mainly for audio tasks, we recommend finetuning the model, while is the goal is a general-purpose assistant with audio capabilities LoRA is a better option.

## F.6  Synthetic data for training

We generated the ReasonAQA synthetic data using the methodology described in Section 2, referring to this generation method as Type 1. However, synthetic data can be created in various ways by modifying prompts and using different language models. To analyze the impact of these factors, we also generated two additional datasets: Type 2 and Type 3.

**Data generation.** In Type 2 data, we modify the prompt while keeping the LLM fixed as Llama 3 8B. The new prompt incorporates expert-designed questions to guide the LLM in generating questions focused on audio and signal reasoning. The Type 2 prompt is shown in Figure 8. For Type 3, we retain the same prompt as Type 2 but upgrade the model from Llama 3 to Llama 3.1 8B. Qualitative examples comparing all three data types for both MCQ and detailed question-answer pairs are presented in Table 16 and Table 17, respectively.

```
{
    "system_prompt": "You are a helpful assistant with expert knowledge about audio, acoustics, and
    ↪    psychoacoustics. Your task is to generation question-answer pairs based on the captions of
    ↪    audio content using natural language.",

    "user_prompt_detail": "You are listening to the audios instead of reading audio captions. You just
    ↪    heard: {caption[i]}. Generate 5 question-answer pairs about what you just heard and return
    ↪    them in JSON format. Please focus on all the events and sounds occurring in the audio clip.
    ↪    Identify and describe each sound source, such as objects, animals, weather, or environmental
    ↪    noises. Include information about the sequence of events and any interactions between sound
    ↪    sources. Make use of the context or setting if it can be inferred from the sounds. Utilize
    ↪    words related to their acoustic properties, such as their semantic relations, their
    ↪    spectro-temporal characteristics, frequency, loudness, duration, materials, interactions, and
    ↪    sound sources. Provide answers in detail. Some example questions to take inspiration from -
    ↪    what is the sound event present in the clip?, what can be inferred about the environment?.
    ↪    Only return the JSON where each dictionary contains a question key and an answer key.",

    "user_prompt_mcq": "You are listening to the audios instead of reading audio captions. You just
    ↪    heard: {caption[i]}. Generate 5 Multiple Choices question-answer pairs about what you just
    ↪    heard and return them in JSON format. Please focus on all the events and sounds occurring in
    ↪    the audio clip. Identify and describe each sound source, such as objects, animals, weather, or
    ↪    environmental noises. Include information about the sequence of events and any interactions
    ↪    between sound sources. Make use of the context or setting if it can be inferred from the
    ↪    sounds. Utilize words related to their acoustic properties, such as their semantic relations,
    ↪    their spectro-temporal characteristics, frequency, loudness, duration, materials,
    ↪    interactions, and sound sources. Provide answers in detail. Some example questions to take
    ↪    inspiration from - what is the sound event present in the clip?, what can be inferred about
    ↪    the environment? Only return the JSON where each dictionary contains question key, choices key
    ↪    with alphabets and an answer key with alphabets."
}
```

Figure 8: LLM system prompt used to generate MCQ and descriptive questions for Type 2 and Type 3 audio-question answer pairs. The "user prompt detail" and "user prompt mcq" shows the prompt used to generate the detailed and MCQ audio question-answer pairs respectively. For Type 2, we use Llama 3 8B as the choice of LLM, while for Type 3, we use Llama 3.1 8B as the LLM. The system, detail and mcq prompts are the same for both Type 2 and Type 3. The results of Type 2 and Type 3 are shown in Table 16 and 17.

**Data analysis.** The statistical analysis of Type 2 and Type 3 data, shown in Table 15, highlights the impact of prompt modifications and model choice on question-answer generation. Type 2 data exhibits lower vocabulary diversity and shorter questions and answers compared to Type 1, indicating that prompt design significantly influences the complexity and richness of the generated question-answer pairs, even when using the same language model. Additionally, Type 3 data shows higher vocabulary diversity and longer questions and answers than Type 2, demonstrating that changes in the LLM can affect the quality of the generated data, independent of the prompt.

| Type | Category | Dataset | # of QA pairs | Question Length (# of words) | Question Vocab. (# of words) | Answer Length (# of words) | Answer Vocab. (# of words) |
|---|---|---|---|---|---|---|---|
| Type 1 | MCQ | AC | 113861 | 11.33 | 7697 | 2.28 | 5411 |
| | | Clotho | 95551 | 11.46 | 8430 | 2.35 | 6009 |
| | | Overall | 209412 | 11.39 | 10035 | 2.31 | 7356 |
| | Detail | AC | 127661 | 3.06 | 3034 | 25.90 | 13181 |
| | | Clotho | 94838 | 2.95 | 3443 | 25.32 | 13445 |
| | | Overall | 222499 | 3.01 | 4295 | 25.65 | 16161 |
| Type 2 | MCQ | AC | 242037 | 13.58 | 9118 | 2.51 | 6600 |
| | | Clotho | 95450 | 13.45 | 8271 | 2.51 | 5950 |
| | | Overall | 337487 | 13.55 | 10805 | 2.51 | 8041 |
| | Detail | AC | 241701 | 3.85 | 3349 | 20.74 | 12302 |
| | | Clotho | 95470 | 3.77 | 3315 | 20.10 | 10794 |
| | | Overall | 337171 | 3.83 | 4453 | 20.56 | 14153 |
| Type 3 | MCQ | AC | 239006 | 15.77 | 9680 | 3.07 | 6984 |
| | | Clotho | 95253 | 15.29 | 8797 | 3.02 | 6325 |
| | | Overall | 334259 | 15.63 | 11408 | 3.06 | 8445 |
| | Detail | AC | 237772 | 4.07 | 3500 | 26.19 | 13488 |
| | | Clotho | 95264 | 4.00 | 3622 | 24.74 | 11811 |
| | | Overall | 333036 | 4.05 | 4764 | 25.77 | 15477 |

Table 15: Statistics of all the generated data types in ReasonAQA

We also explore the qualitative differences in the generated questions and responses between these datasets (Type 2 and Type 3) and the original dataset (Type 1). To do this, we identify words that appear in Type 2 but not in Type 1 to determine unique audio characteristics present in Type 2. Similarly, the same method is applied to examine the distinctions between Type 3 and Type 1. The results of this analysis, shown in Figure 9, reveal notable shifts in word usage, highlighting differences in question formulation and response patterns. Specifically, Type 2 and Type 3 questions show an increased occurrence of words such as environment, inferred, and setting, suggesting a stronger emphasis on reasoning about environmental and contextual aspects of the audio. This aligns with the prompt instructions, which include directives such as "focus on all the events and sounds occurring in the audio clip" and "make use of the context or setting if it can be inferred from the sounds." Similarly, answers in these datasets contain words like likely, possibly, and suggests, indicating a greater focus on inference and probabilistic reasoning compared to Type 1. Another key finding is that despite differences in the underlying LLMs, Type 2 and Type 3 data exhibit similar vocabulary patterns, with certain words appearing more frequently in both compared to Type 1. This consistency underscores the importance of prompt design in shaping the generated data, demonstrating that the structure of the prompt significantly influences the nature of the questions and answers, even when different LLMs are used.

### F.7 Scaling audio data

In Mellow, our primary objective was to enhance the model's inherent reasoning ability through architectural, learning, or post-training techniques rather than by increasing data (audio) coverage—such as expanding knowledge of audio objects and scenes. A natural follow-up question arises: *now that we have a recipe for training small audio-language models, does performance improve when we scale the training data?* To investigate this, we leverage the WavCaps dataset [57]. Using the same method described in Section F.6, we generate approximately 1.80 million detailed QA pairs and 1.82 million multiple-choice QA pairs for WavCaps, resulting in a total of 3.6 million QA pairs. For our experiment, we incorporate these WavCaps QA pairs into the ReasonAQA type 1 dataset.

The results, presented in Table 14, reveal an absolute 1.5% improvement in overall MMAU performance. Notably, music-related performance increases by an absolute 10%, likely due to WavCaps' heavy reliance on FreeSound (70%), which consists predominantly of music data. Performance on MMAU test-mini sound remains comparable, while speech performance appears random since we do not train on speech data. In the case of Audio Captioning, we observe a decline in performance. A closer analysis of the model's output suggests that this drop is primarily due to limitations in the SPICE metric, as discussed in Section 18. Meanwhile, for Audio Entailment, we see a slight improvement, which can be attributed to the model's improved understanding of audio concepts absent in the type 1 dataset, allowing the use of learned reasoning chains over this new knowledge. Overall, these results demonstrate that expanding the range of audio concepts in the training data

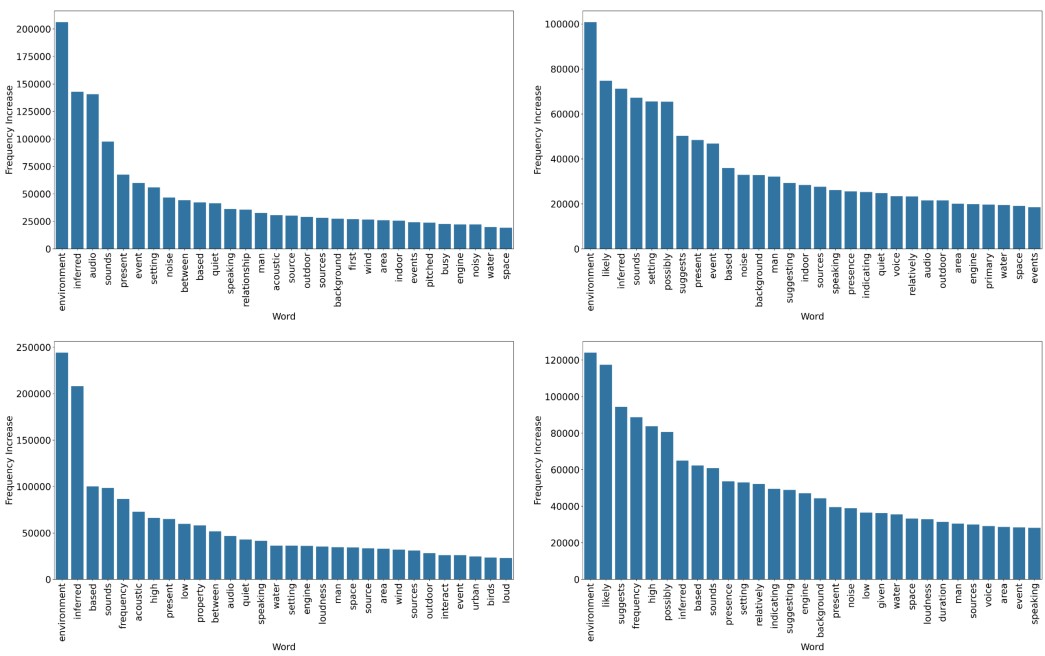

Figure 9: Comparison of most frequent words in type 3 questions and answers that are not present in type 1. Comparison of most frequent words in type 2 questions (top left), type 2 answers (top right), type 3 questions (bottom left) and type 3 answers (bottom right) that are not present in type 1

leads to improved model performance. Future work could explore scaling laws for audio, allowing extrapolation of small-scale experiments to predict the effects of training on significantly larger audio datasets.

# G    Metric SPICE

SPICE [3] is sensitive to wording variations and proves overly stringent for evaluating the audio question-answering task. As shown in Table 18, SPICE performs well when the predicted response exactly matches the reference (e.g., "reading" and "reading") or is very similar (e.g., "bus" and "yes, it is a bus"). However, when the predicted response is longer or phrased differently from the reference, SPICE tends to assign low scores even to correct answers. As responses become more complex, SPICE frequently assigns a zero score, failing to recognize their correctness. To address these limitations, we also incorporate the latest metrics from the audio-language literature [21, 22], such as the AURA metric, to evaluate the model's performance on the audio question-answering task.

# H    Instruction following

Audio-Language Models (ALMs) are capable of performing various tasks, including answering open-ended questions. However, many ALMs in the literature are known to struggle with following instructions when these instructions deviate significantly from their training distribution. This issue is particularly evident when answering deductive reasoning questions [18] or multiple-choice questions (MCQs) [63], where the model might only reply with a yes or no, select one of the options arbitrarily, or provide an entirely unrelated answer. In our experiments, we observed that Mellow exhibits similar issues with instruction following.

To evaluate instruction following, we selected the MMAU benchmark as our target. Although Mellow can handle MCQ tasks, the audio, questions, and answers in MMAU are out-of-distribution relative to Mellow's training data. This setup allowed us to assess how often Mellow fails to select an appropriate option in the MCQ format, instead producing irrelevant responses that do not adhere to the provided options or the question. Out of approximately 10k questions in MMAU, we found that Mellow failed to pick an appropriate option in roughly 20% of the cases. The reasons for this failure can be categorized as follows:

| |
|---|
| **Caption: A man talks nearby as water pours out of a faucet** |

*Type 1*

**Question:** What is the primary sound event present in the clip? a) A person speaking b) Water pouring from a faucet c) Background noise d) A combination of both
**Answer:** d) A combination of both

**Question:** What is the most likely material of the object making the sound? a) Metal b) Glass c) Plastic d) Ceramic
**Answer:** a) Metal

**Question:** What is the approximate loudness of the sound? a) Very loud b) Moderately loud c) Soft d) Very soft
**Answer:** b) Moderately loud

**Question:** What is the most similar sound to the sound of water pouring from a faucet? a) Rainfall b) Ocean waves c) Running water d) A babbling brook
**Answer:** c) Running water

**Question:** What is the purpose of the sound of water pouring from a faucet? a) To create a background noise b) To mask other sounds c) To create a sense of activity d) To create a sense of calm
**Answer:** c) To create a sense of activity

*Type 2*

**Question:** What is the primary sound event present in the clip? A) A man talking B) Background noise C) Other D) Water pouring out of a faucet
**Answer:** D) Water pouring out of a faucet

**Question:** What is the sound source of the pouring water? A) A stream B) A bucket C) A pool D) A faucet
**Answer:** D) A faucet

**Question:** What can be inferred about the environment? A) It is an outdoor setting B) It is a private place C) It is a public place D) It is an indoor setting
**Answer:** D) It is an indoor setting

**Question:** What is the relationship between the man's voice and the pouring water? A) The man's voice is louder than the water B) The man's voice is simultaneous with the water C) The man's voice is softer than the water D) The man's voice is before the water
**Answer:** B) The man's voice is simultaneous with the water

**Question:** What is the dominant acoustic property of the sound of the pouring water? A) Frequency B) Duration C) Material D) Loudness
**Answer:** D) Loudness

*Type 3*

**Question:** What is the sound event present in the clip? A) A person talking B) Neither A nor B C) Water pouring out of a faucet D) Both A and B
**Answer:** D) Both A and B

**Question:** What can be inferred about the environment? A) It is in a public place, such as a restaurant or store B) It is in a quiet, isolated area C) It is outdoors, near a stream or river D) It is indoors, likely in a kitchen or bathroom
**Answer:** D) It is indoors, likely in a kitchen or bathroom

**Question:** What is the material of the object making the sound of water pouring? A) Metal B) Glass C) Plastic D) Wood
**Answer:** A) Metal

**Question:** What is the semantic relation between the sound of the man talking and the sound of water pouring? A) They are competing for attention B) The sound of the water pouring is causing the man to talk C) They are occurring simultaneously and independently D) The sound of the man talking is causing the water to pour
**Answer:** C) They are occurring simultaneously and independently

**Question:** What can be inferred about the loudness of the sound of the man talking? A) It is much louder than the sound of water pouring B) It is roughly equal in loudness to the sound of water pouring C) It is much softer than the sound of water pouring D) It is slightly louder than the sound of water pouring
**Answer:** D) It is slightly louder than the sound of water pouring

Table 16: Comparing ReasonAQA Type 1, Type 2 and Type 3 examples. From both datasets, we sample MCQ questions generated for the same audio event.

- **Lack of task-specific knowledge.** In cases such as music chord identification or phoneme recognition, Mellow does not possess the necessary knowledge to identify individual chord letters or phonemes. As a result, the model produces responses based on related knowledge (e.g., referencing guitar playing) instead of selecting an option from the MCQ list.

- **Out-of-distribution symbols.** When the options contain symbols that were not present in the training data (for example, an underscore or a differently formatted list of answers), the model sometimes generates gibberish or random responses. Ideally, the model should still select the correct option or produce a valid English response but fails to do so.

- **String parsing errors.** Occasionally, the model produces a response that is a valid option but includes extraneous characters, such as an extra space, exclamation mark, or question mark. These errors can cause traditional string-parsing methods to fail. This issue could be mitigated by using more robust evaluation metrics, such as leveraging a language model for parsing instead of relying solely on string matching.

Overall, similar to other existing ALMs, Mellow's ability to follow instructions has room for improvement. Some of these errors could be resolved by increasing the training data and incorporating a more diverse set of instructions and answer distributions. However, it is also possible that at compute optimal frontier, larger models are inherently better suited to generalize to novel instructions compared to smaller models.

| |
|---|
| **Caption: A man talks nearby as water pours out of a faucet** |

*Type 1*

**Question:** What is the primary sound event present in the clip?

**Answer:** The primary sound event present in the clip is the sound of a man talking, accompanied by the sound of water pouring out of a faucet. The talking is a dynamic sound with varying pitch, tone, and volume, while the water pouring sound is a steady, continuous sound with a consistent pitch and volume.

**Question:** What is the acoustic property of the water pouring sound?

**Answer:** The water pouring sound has a characteristic of a steady, continuous flow, which is likely due to the consistent flow rate of the water. This sound also has a gentle, soothing quality, which is often associated with a low to moderate frequency range (around 100-500 Hz) and a moderate to high volume level.

**Question:** How would you describe the tone of the man's voice?

**Answer:** The tone of the man's voice is likely conversational and casual, with a natural, relaxed pitch and a moderate volume level. The tone may also convey a sense of familiarity or routine, as the man is likely speaking in an everyday conversation rather than in a formal or dramatic context.

**Question:** What is the sound most similar to?

**Answer:** The sound of the man talking and the water pouring is most similar to a typical morning routine, such as brushing teeth or washing hands. The combination of the gentle water sound and the conversational tone creates a sense of familiarity and comfort.

**Question:** What is the psychoacoustic effect of this sound?

**Answer:** The sound of the man talking and the water pouring may have a calming or soothing effect on the listener, as the gentle water sound and the conversational tone create a sense of relaxation and familiarity. The sound may also help to distract from other stimuli or create a sense of focus, as the listener is drawn to the gentle, continuous flow of the water.

*Type 2*

**Question:** What sound event is present in the clip?

**Answer:** The sound event present in the clip is a man talking and water pouring out of a faucet. The talking is a low-pitched, steady-state sound with a moderate loudness, while the water pouring is a high-pitched, transient sound with varying loudness.

**Question:** What can be inferred about the environment?

**Answer:** From the sound of the water pouring out of a faucet, it can be inferred that the environment is likely a indoor setting, such as a kitchen or bathroom, with a sink and running water. The man's talking suggests that he is likely in the same room as the listener.

**Question:** What is the acoustic property of the man's talking?

**Answer:** The man's talking has a low-pitched, steady-state quality, which suggests that his voice is likely deep and resonant. The moderate loudness of his talking indicates that he is speaking at a normal volume, not too loud or too soft.

**Question:** What is the acoustic property of the water pouring?

**Answer:** The water pouring has a high-pitched, transient quality, which suggests that it is a fast-moving fluid with a lot of energy. The varying loudness of the water pouring indicates that the flow rate of the water is changing, possibly due to the man adjusting the faucet.

**Question:** What is the interaction between the sound sources?

**Answer:** The man's talking and the water pouring are two separate sound sources that are occurring simultaneously. The talking is a steady-state sound, while the water pouring is a transient sound, which suggests that the man is likely talking while he is adjusting the faucet or observing the water pouring. There is no apparent interaction between the two sound sources, but they coexist in the same environment.

*Type 3*

**Question:** What is the primary sound event present in the clip?

**Answer:** The primary sound event present in the clip is a man talking nearby, accompanied by the sound of water pouring out of a faucet.

**Question:** What can be inferred about the environment based on the sounds?

**Answer:** Based on the sounds, it can be inferred that the environment is likely a domestic or residential setting, such as a kitchen or bathroom, where a faucet is present and a person is nearby.

**Question:** What are the acoustic properties of the sound of water pouring out of a faucet?

**Answer:** The sound of water pouring out of a faucet is a continuous, flowing sound with a steady, consistent pitch and a moderate to high loudness level. It is likely a broadband noise with a frequency range of around 20 Hz to 20 kHz, and a duration that is sustained throughout the clip.

**Question:** How do the sound of the man talking and the sound of water pouring interact?

**Answer:** The sound of the man talking and the sound of water pouring interact by creating a sense of spatial separation between the two sound sources. The talking sound is likely coming from a closer location to the microphone, while the water pouring sound is coming from a farther location, creating a sense of depth and distance between the two sound sources.

**Question:** What can be inferred about the man's proximity to the microphone based on the sound of his voice?

**Answer:** Based on the sound of his voice, it can be inferred that the man is likely standing relatively close to the microphone, as his voice is loud and clear, with minimal reverberation or echo. This suggests that he is in a relatively small, enclosed space, such as a kitchen or bathroom, where the sound of his voice can be easily heard.

Table 17: Comparing ReasonAQA Type 1, Type 2 and Type 3 examples. From both datasets, we sample detail questions generated for the same audio event.

# I  Hallucination

Language models often struggle with hallucination, generating outputs that are misleading or incorrect. For Audio-Language Models (ALMs), this issue manifests as producing responses not based on actual audio input. For instance, an ALM might invent audio events, misinterpret common sounds like human speech, or completely disregard audio information. This lack of grounding can impair the model's ability to answer user's questions faithfully. Therefore, we take inspiration from past work [19], and provide users with a way to detect hallucinations. To achieve this, we use the HTSAT audio encoder, which is part of Mellow. It is trained on AudioSet, and can predict 527 audio events and their presence probabilities over time. This enables users to view the detected audio events and their presence across time. Users can then compare the generated descriptions with the audio event presence probabilities to verify the description's accuracy. An example of this is shown in Figure

| Question | Reference Answer | Answer | Human | SPICE | AURA |
|---|---|---|---|---|---|
| What activity is taking place? | reading | reading | 1.00 | 1.00 | 1.00 |
| Can you identify the type of vehicle in the audio clip? | Yes, it is a bus. | bus | 1.00 | 1.00 | 1.00 |
| What kind of brush is the person using? | teeth | Based on the sound, it sounds like the person is using a toothbrush to brush their teeth. | 1.00 | 0.22 | 1.00 |
| What liquid flowing sound is this? | water | This is the sound of flowing water. | 1.00 | 0.40 | 1.00 |
| Is someone using a machine? | yes | Yes, a machine is running. | 1.00 | 0.00 | 1.00 |
| In what room would these items likely be found? | kitchen | These items are typically found in a kitchen or dining area, where glasses and silverware are commonly used. | 1.00 | 0.00 | 1.00 |

Table 18: AURA scores correlate better than SPICE with human scores on open-ended question answers. GitHub link: https://github.com/satvik-dixit/aura

10. The top pane displays the audio difference explanation generated by the model. For each audio, the log mel spectrogram and the top three audio event presence probabilities over time are plotted. From the figure, we see the detected audio events and compare them to the generated description from Mellow. Though this method requires human-in-loop and is not fully automated, it provides a way to detect hallucinations in critical scenarios.

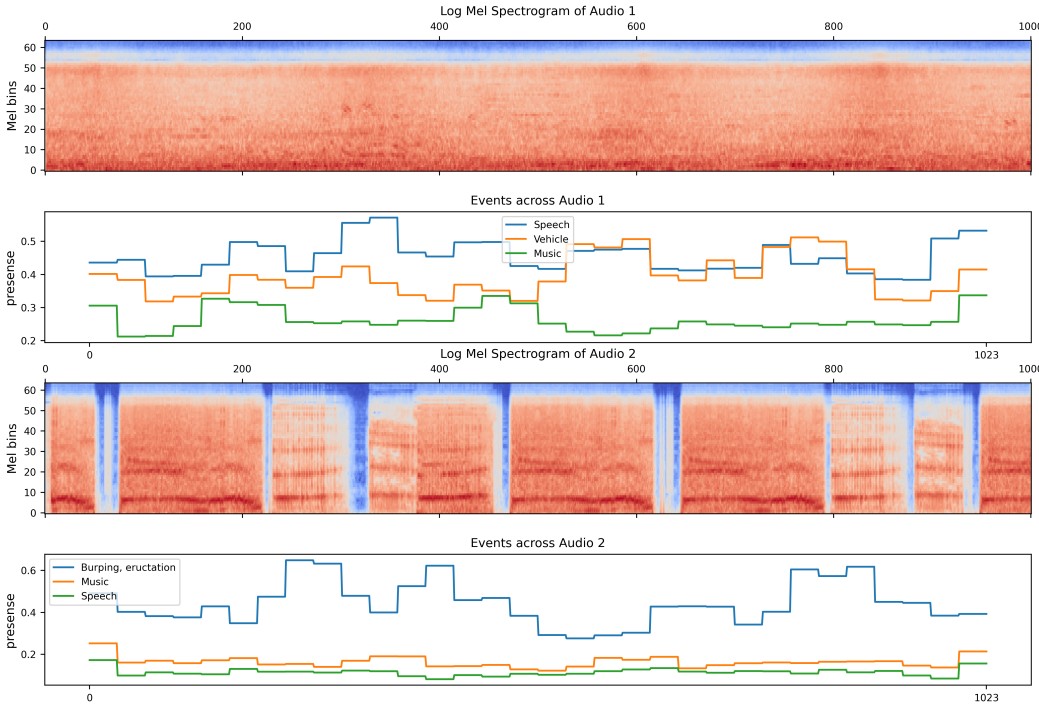

Figure 10: The user can use audio event presence probabilities from HTSAT to detect potential hallucinations in the language model output of Mellow.

## J   Limitations

Despite Mellow's strong performance on multiple downstream tasks, it has several limitations. First, Mellow is not a general-purpose audio model. While it can understand sound, music, and paralinguistics to a certain extent, it is not trained for speech-content tasks such as transcription or translation. By training on limited data, Mellow has limited knowledge of audio concepts, which can be improved with scaling training pairs. Second, in Mellow, we fine-tune the language model

| Model | Response |
|---|---|

**Non-binary questions**

*Question: What is the helicopter doing? Reference Answer: flying*

| | |
|---|---|
| Audio Flamingo | flying |
| GAMA | The helicopter seems to be hovering or stationary, as indicated by the constant presence of its sound throughout the audio. |
| Qwen Audio Chat | The helicopter is flying. |
| Qwen 2 Audio | The helicopter is flying overhead. |
| **Mellow** | The helicopter is flying in the background, creating a constant whooshing or whirring sound that is audible throughout the clip. The sound is likely due to the rotor blades of the helicopter moving through the air. |

*Question: What is on fire? Reference Answer: fireplace*

| | |
|---|---|
| Audio Flamingo | brake |
| GAMA | The audio does not provide specific information about what's on fire, but it could be a campfire or a fireplace in the context of a home setting during a rainstorm. |
| Qwen Audio Chat | It sounds like rain falling on various surfaces. |
| Qwen 2 Audio | A pot of something is on fire. |
| **Mellow** | The fire is burning wood, which is evident from the crackling and popping sounds heard in the clip. The wood is likely dry and combustible, given the intensity and frequency of the crackling and popping sounds. |

*Question: What is the gender of the person speaking? Reference Answer: male*

| | |
|---|---|
| Audio Flamingo | male |
| GAMA | The speaker is male, as indicated by the audio label and the presence of a man's voice in the recording. |
| Qwen Audio Chat | The gender of the person speaking is male. |
| Qwen 2 Audio | The speaker is male. |
| **Mellow** | the gender of the person speaking is an adult male, as inferred from the deep and resonant tone of his voice. |

**Binary questions**

*Question: Are the children playing outside? Reference Answer: yes*

| | |
|---|---|
| Audio Flamingo | yes |
| GAMA | Yes, the children are likely playing outside based on the continuous presence of their voices and the sounds of birds chirping in the background. |
| Qwen Audio Chat | Yes, the children are playing outside. |
| Qwen 2 Audio | Yes |
| **Mellow** | yes, the children are playing outside, as evidenced by the sounds of children's laughter, footsteps, and possibly some toys or games being played. the sounds of children playing outside are likely to be a combination of footsteps, laughter, and possibly some toys being moved around. |

*Question: Is this taking place outside? Reference Answer: yes*

| | |
|---|---|
| Audio Flamingo | yes |
| GAMA | The presence of wind sounds suggests that the scene is likely outdoors, possibly near a road. |
| Qwen Audio Chat | Yes, this is taking place outside. |
| Qwen 2 Audio | Yes |
| **Mellow** | yes, this sound is taking place outside, as the bus is moving and the sound is carrying over a distance. the sound is not muffled or dampened, indicating that the bus is not enclosed in a building or heavily vegetated area. |

*Question: Are the men swimming? Reference Answer: no*

| | |
|---|---|
| Audio Flamingo | no |
| GAMA | No, the sounds suggest that they are likely in a vehicle or on a bus, as indicated by the engine and impact noises. |
| Qwen Audio Chat | No, the men are not swimming. They are working on a bus or a truck. |
| Qwen 2 Audio | No, the men are not swimming. |
| **Mellow** | no, the men are not swimming in the clip. the audio clip appears to be focused solely on the sound of the truck driving away, with no additional sounds or ambient noise present. |

Table 19: Example of different Audio-Language Model responses to audio-based questions from ClothoAQA

component. This causes the model to forget factual knowledge unrelated to audio, such as information about the capitals of specific countries or historical and present-day facts that are not relevant to audio. Third, Mellow is not a chat-based model and cannot be used as a general-purpose assistant like some existing models in the literature [10, 11]. Instead, Mellow closely follows the instruction-tuning paradigm seen in models like T5 [61], FLAN [72], SigLIP2 [70], and PaliGemma series [5, 64] where the goal is to enable a model to perform multiple perception tasks through language. Fourth, as the data scale increases, Mellow's parameter count will likely need to be scaled as well to maintain a compute-optimal frontier. However, with Mellow, we aim to demonstrate that the compute-optimal frontier for speech and audio tasks may be significantly lower than previously estimated. In summary, we hope this work inspires researchers to explore small audio-language models capable of performing multiple tasks, alongside the existing research on general-purpose audio assistants.

**Reasoning over context**

**Q: What can be inferred about the materials involved in the sound?**
M: From the sounds, it can be inferred that the toilet bowl and pipe are likely made of ceramic or porcelain, as these materials produce a distinctive sound when water flows through them. The sound of the water draining also suggests that the pipe is likely made of a smooth material, such as PVC or ABS, which produces a low-frequency rumble when water flows through it.

**Q: What can be inferred from this audio?**
M: From this audio, you can infer that the woman is likely in a social setting, possibly with a friend or family member, and that she is enjoying the conversation or sharing her thoughts with someone else. The laughter and conversation suggest a lighthearted and playful atmosphere.

**Q: What can be inferred about the environment?**
M: The environment is likely a natural setting, such as a pond or lake, with a body of water, vegetation, and possibly some buildings or structures in the background, based on the sounds of water splashing and rustling.

**Comparative Reasoning**

**Q: Explain the difference between audios in short**
M: The two audios differ significantly in their acoustic properties, with the first audio featuring a chaotic mixture of high-frequency buzzing sounds with varying loudness and duration, whereas the second audio has a structured soundscape with a low-frequency rumble of the vehicle, punctuated by the clear and distinct voices of the people speaking.

**Q: Explain the difference between the two audios in detail**
M: As I listen to the two audios, I notice distinct differences in their acoustic properties. The first audio, featuring a kid screaming at a park, has a sharp, high-pitched frequency with a rising loudness level, indicating a sudden emotional outburst. The background noise of people chatting and laughing creates a gentle hum, adding a sense of an environment context. In contrast, the second audio is a truck driving down the street, has a deep, rumbling frequency with a steady, constant loudness level, characteristic of a powerful engine. The background noise of traffic, including car horns and revving engines, creates a cacophonous soundscape, evoking a sense of urban chaos. While both audios have high levels of loudness, the kid's scream is more attention-grabbing, whereas the truck's engine is sustained.

**Identifying and reasoning over sound events**

**Q: How do the two sound events interact with each other?**
M: The baby's crying sound and the TV sound interact with each other in a way that the crying sound dominates the auditory scene. The TV sound is present in the background, but it is masked by the loudness of the crying sound. This suggests that the caregiver or family member is trying to create a distraction or background noise to soothe the baby.

**Q: How do different sounds interact with each other?**
M: The engine sound is the dominant sound in the clip, and it appears to be the primary focus of attention. the high-pitched whining sound is secondary and adds a sense of tension or urgency to the overall sound. The interaction between the two sounds creates a sense of dynamic energy and movement, suggesting that the engine is in motion or under stress.

**Q: How do the duck calls relate to the other sound events in the clip?**
M: The duck calls are likely being blown by a person, and are used to interact with the ducks, possibly to lure them or to communicate with them, and are heard in conjunction with the human conversation, suggesting a scene of duck hunting or observation.

**Q: What happens to the stirring sound over time?**
M: The stirring sound gradually slows down and becomes less intense, with a decrease in frequency and loudness. The spoon's motion becomes more deliberate and controlled, resulting in a lower amplitude sound with a frequency range of approximately 100-400 Hz and a loudness of around 40-50 dB. The sound transitions from a sharp, percussive sound to a softer, more muffled sound.

**Q: Can you identify the specific type of aircraft engine from the audio?**
M: While it's difficult to pinpoint the exact type of engine, the sound has a characteristic 'whoosh' or 'whine' quality, which is typical of jet engines. the engine's pitch and tone are consistent with a high-performance jet engine, possibly a turbojet or a turboprop engine.

**Deductive Reasoning**

**Q: Based on the audio, what can be said about the hypothesis - "A farmer is giving a tour of his ranch while chickens roam nearby"? a) It is definitely true b) It is definitely false c) It is plausible d) I cannot determine**
M: b) It is definitely false

**Multiple Choice Questions**

**Q: What is the acoustic property of the wind sound? A) Soft and gentle B) Low frequency and rumbling C) High frequency and loud D) Sharp and piercing**
M. A) Soft and gentle

**Q: What is the characteristic of the machine's sound? A) Whiny and squealing B) High-pitched and smooth C) Low-pitched and rough D) Silent**
M: A) Whiny and squealing

**Q: What is the quality of the audio in the clip? a) High-quality b) Medium-quality c) Low-quality d) Excellent-quality**
M: c) Low-quality

**Q: What is the primary sound event present in the clip? a) vacuum cleaner b) chirping birds c) car horn d) clapping**
M: vacuum cleaner

**Binary Questions**

**Q: Is there more than one bird squawking in the audio clip? Answer only using yes or no.**
M: yes

Figure 11: Mellow's different capabilities and examples.


Figure 12: Mellow is able to reason over complicated audio events and generate detailed answers as shown on left. Mellow also has the ability to recognize information that cannot be inferred from the audio alone – such as the origin of the siren – as shown on right.

- The claims made should match theoretical and experimental results, and reflect how much the results can be expected to generalize to other settings.
- It is fine to include aspirational goals as motivation as long as it is clear that these goals are not attained by the paper.

2. **Limitations**

Question: Does the paper discuss the limitations of the work performed by the authors?

Answer: [Yes]

Justification: Yes, the limitations are highlighted in Section J in Appendix.

Guidelines:

- The answer NA means that the paper has no limitation while the answer No means that the paper has limitations, but those are not discussed in the paper.
- The authors are encouraged to create a separate "Limitations" section in their paper.
- The paper should point out any strong assumptions and how robust the results are to violations of these assumptions (e.g., independence assumptions, noiseless settings, model well-specification, asymptotic approximations only holding locally). The authors should reflect on how these assumptions might be violated in practice and what the implications would be.
- The authors should reflect on the scope of the claims made, e.g., if the approach was only tested on a few datasets or with a few runs. In general, empirical results often depend on implicit assumptions, which should be articulated.
- The authors should reflect on the factors that influence the performance of the approach. For example, a facial recognition algorithm may perform poorly when image resolution is low or images are taken in low lighting. Or a speech-to-text system might not be used reliably to provide closed captions for online lectures because it fails to handle technical jargon.
- The authors should discuss the computational efficiency of the proposed algorithms and how they scale with dataset size.
- If applicable, the authors should discuss possible limitations of their approach to address problems of privacy and fairness.
- While the authors might fear that complete honesty about limitations might be used by reviewers as grounds for rejection, a worse outcome might be that reviewers discover limitations that aren't acknowledged in the paper. The authors should use their best judgment and recognize that individual actions in favor of transparency play an important role in developing norms that preserve the integrity of the community. Reviewers will be specifically instructed to not penalize honesty concerning limitations.

3. **Theory assumptions and proofs**

Question: For each theoretical result, does the paper provide the full set of assumptions and a complete (and correct) proof?

Answer: [NA]

Justification: The manuscript contains experimental results and not new proofs.

Guidelines:

- The answer NA means that the paper does not include theoretical results.
- All the theorems, formulas, and proofs in the paper should be numbered and cross-referenced.
- All assumptions should be clearly stated or referenced in the statement of any theorems.
- The proofs can either appear in the main paper or the supplemental material, but if they appear in the supplemental material, the authors are encouraged to provide a short proof sketch to provide intuition.
- Inversely, any informal proof provided in the core of the paper should be complemented by formal proofs provided in appendix or supplemental material.
- Theorems and Lemmas that the proof relies upon should be properly referenced.

4. **Experimental result reproducibility**

Question: Does the paper fully disclose all the information needed to reproduce the main experimental results of the paper to the extent that it affects the main claims and/or conclusions of the paper (regardless of whether the code and data are provided or not)?

Answer: [Yes]

Justification: In every result section, we describe the dataset and experimental setup used.

Guidelines:

- The answer NA means that the paper does not include experiments.
- If the paper includes experiments, a No answer to this question will not be perceived well by the reviewers: Making the paper reproducible is important, regardless of whether the code and data are provided or not.
- If the contribution is a dataset and/or model, the authors should describe the steps taken to make their results reproducible or verifiable.
- Depending on the contribution, reproducibility can be accomplished in various ways. For example, if the contribution is a novel architecture, describing the architecture fully might suffice, or if the contribution is a specific model and empirical evaluation, it may be necessary to either make it possible for others to replicate the model with the same dataset, or provide access to the model. In general. releasing code and data is often one good way to accomplish this, but reproducibility can also be provided via detailed instructions for how to replicate the results, access to a hosted model (e.g., in the case of a large language model), releasing of a model checkpoint, or other means that are appropriate to the research performed.
- While NeurIPS does not require releasing code, the conference does require all submissions to provide some reasonable avenue for reproducibility, which may depend on the nature of the contribution. For example
  (a) If the contribution is primarily a new algorithm, the paper should make it clear how to reproduce that algorithm.
  (b) If the contribution is primarily a new model architecture, the paper should describe the architecture clearly and fully.
  (c) If the contribution is a new model (e.g., a large language model), then there should either be a way to access this model for reproducing the results or a way to reproduce the model (e.g., with an open-source dataset or instructions for how to construct the dataset).
  (d) We recognize that reproducibility may be tricky in some cases, in which case authors are welcome to describe the particular way they provide for reproducibility. In the case of closed-source models, it may be that access to the model is limited in some way (e.g., to registered users), but it should be possible for other researchers to have some path to reproducing or verifying the results.

5. **Open access to data and code**

   Question: Does the paper provide open access to the data and code, with sufficient instructions to faithfully reproduce the main experimental results, as described in supplemental material?

   Answer: [Yes]

   Justification: Yes, we released the ReasonAQA dataset and will subsequently release the training and evaluation code in June to reproduce all experiments.

   Guidelines:

   - The answer NA means that paper does not include experiments requiring code.
   - Please see the NeurIPS code and data submission guidelines (`https://nips.cc/public/guides/CodeSubmissionPolicy`) for more details.
   - While we encourage the release of code and data, we understand that this might not be possible, so "No" is an acceptable answer. Papers cannot be rejected simply for not including code, unless this is central to the contribution (e.g., for a new open-source benchmark).
   - The instructions should contain the exact command and environment needed to run to reproduce the results. See the NeurIPS code and data submission guidelines (`https://nips.cc/public/guides/CodeSubmissionPolicy`) for more details.
   - The authors should provide instructions on data access and preparation, including how to access the raw data, preprocessed data, intermediate data, and generated data, etc.
   - The authors should provide scripts to reproduce all experimental results for the new proposed method and baselines. If only a subset of experiments are reproducible, they should state which ones are omitted from the script and why.
   - At submission time, to preserve anonymity, the authors should release anonymized versions (if applicable).
   - Providing as much information as possible in supplemental material (appended to the paper) is recommended, but including URLs to data and code is permitted.

6. **Experimental setting/details**

   Question: Does the paper specify all the training and test details (e.g., data splits, hyperparameters, how they were chosen, type of optimizer, etc.) necessary to understand the results?

   Answer: [Yes]

   Justification: Yes, all experimental details are available in Section E in Appendix.

   Guidelines:

   - The answer NA means that the paper does not include experiments.
   - The experimental setting should be presented in the core of the paper to a level of detail that is necessary to appreciate the results and make sense of them.
   - The full details can be provided either with the code, in appendix, or as supplemental material.

7. **Experiment statistical significance**

   Question: Does the paper report error bars suitably and correctly defined or other appropriate information about the statistical significance of the experiments?

   Answer: [Yes]

   Justification: The experimental results are averaged over 5 inference runs of the model.

   Guidelines:

   - The answer NA means that the paper does not include experiments.
   - The authors should answer "Yes" if the results are accompanied by error bars, confidence intervals, or statistical significance tests, at least for the experiments that support the main claims of the paper.
   - The factors of variability that the error bars are capturing should be clearly stated (for example, train/test split, initialization, random drawing of some parameter, or overall run with given experimental conditions).

- The method for calculating the error bars should be explained (closed form formula, call to a library function, bootstrap, etc.)
- The assumptions made should be given (e.g., Normally distributed errors).
- It should be clear whether the error bar is the standard deviation or the standard error of the mean.
- It is OK to report 1-sigma error bars, but one should state it. The authors should preferably report a 2-sigma error bar than state that they have a 96% CI, if the hypothesis of Normality of errors is not verified.
- For asymmetric distributions, the authors should be careful not to show in tables or figures symmetric error bars that would yield results that are out of range (e.g. negative error rates).
- If error bars are reported in tables or plots, The authors should explain in the text how they were calculated and reference the corresponding figures or tables in the text.

8. **Experiments compute resources**

   Question: For each experiment, does the paper provide sufficient information on the computer resources (type of compute workers, memory, time of execution) needed to reproduce the experiments?

   Answer: [Yes]

   Justification: Yes, the details are available in Appendix E.

   Guidelines:
   - The answer NA means that the paper does not include experiments.
   - The paper should indicate the type of compute workers CPU or GPU, internal cluster, or cloud provider, including relevant memory and storage.
   - The paper should provide the amount of compute required for each of the individual experimental runs as well as estimate the total compute.
   - The paper should disclose whether the full research project required more compute than the experiments reported in the paper (e.g., preliminary or failed experiments that didn't make it into the paper).

9. **Code of ethics**

   Question: Does the research conducted in the paper conform, in every respect, with the NeurIPS Code of Ethics https://neurips.cc/public/EthicsGuidelines?

   Answer: [Yes]

   Justification: Yes, the manuscript and work complies with the NeurIPS code of ethics.

   Guidelines:
   - The answer NA means that the authors have not reviewed the NeurIPS Code of Ethics.
   - If the authors answer No, they should explain the special circumstances that require a deviation from the Code of Ethics.
   - The authors should make sure to preserve anonymity (e.g., if there is a special consideration due to laws or regulations in their jurisdiction).

10. **Broader impacts**

    Question: Does the paper discuss both potential positive societal impacts and negative societal impacts of the work performed?

    Answer: [Yes]

    Justification: Yes, we acknowledge this in Section A in Appendix.

    Guidelines:
    - The answer NA means that there is no societal impact of the work performed.
    - If the authors answer NA or No, they should explain why their work has no societal impact or why the paper does not address societal impact.
    - Examples of negative societal impacts include potential malicious or unintended uses (e.g., disinformation, generating fake profiles, surveillance), fairness considerations (e.g., deployment of technologies that could make decisions that unfairly impact specific groups), privacy considerations, and security considerations.

- The conference expects that many papers will be foundational research and not tied to particular applications, let alone deployments. However, if there is a direct path to any negative applications, the authors should point it out. For example, it is legitimate to point out that an improvement in the quality of generative models could be used to generate deepfakes for disinformation. On the other hand, it is not needed to point out that a generic algorithm for optimizing neural networks could enable people to train models that generate Deepfakes faster.
- The authors should consider possible harms that could arise when the technology is being used as intended and functioning correctly, harms that could arise when the technology is being used as intended but gives incorrect results, and harms following from (intentional or unintentional) misuse of the technology.
- If there are negative societal impacts, the authors could also discuss possible mitigation strategies (e.g., gated release of models, providing defenses in addition to attacks, mechanisms for monitoring misuse, mechanisms to monitor how a system learns from feedback over time, improving the efficiency and accessibility of ML).

11. **Safeguards**

Question: Does the paper describe safeguards that have been put in place for responsible release of data or models that have a high risk for misuse (e.g., pretrained language models, image generators, or scraped datasets)?

Answer: [Yes]

Justification: We manually filter the data to remove any potential harmful examples related to gunshot, violence and surveillance.

Guidelines:

- The answer NA means that the paper poses no such risks.
- Released models that have a high risk for misuse or dual-use should be released with necessary safeguards to allow for controlled use of the model, for example by requiring that users adhere to usage guidelines or restrictions to access the model or implementing safety filters.
- Datasets that have been scraped from the Internet could pose safety risks. The authors should describe how they avoided releasing unsafe images.
- We recognize that providing effective safeguards is challenging, and many papers do not require this, but we encourage authors to take this into account and make a best faith effort.

12. **Licenses for existing assets**

Question: Are the creators or original owners of assets (e.g., code, data, models), used in the paper, properly credited and are the license and terms of use explicitly mentioned and properly respected?

Answer: [Yes]

Justification: Yes, we use two audio datasets to create ReasonAQA and the dataset and creators are properly credited.

Guidelines:

- The answer NA means that the paper does not use existing assets.
- The authors should cite the original paper that produced the code package or dataset.
- The authors should state which version of the asset is used and, if possible, include a URL.
- The name of the license (e.g., CC-BY 4.0) should be included for each asset.
- For scraped data from a particular source (e.g., website), the copyright and terms of service of that source should be provided.
- If assets are released, the license, copyright information, and terms of use in the package should be provided. For popular datasets, `paperswithcode.com/datasets` has curated licenses for some datasets. Their licensing guide can help determine the license of a dataset.

- For existing datasets that are re-packaged, both the original license and the license of the derived asset (if it has changed) should be provided.
- If this information is not available online, the authors are encouraged to reach out to the asset's creators.

13. **New assets**

Question: Are new assets introduced in the paper well documented and is the documentation provided alongside the assets?

Answer: [Yes]

Justification: Yes, we provide data statistics and model checkpoints.

Guidelines:

- The answer NA means that the paper does not release new assets.
- Researchers should communicate the details of the dataset/code/model as part of their submissions via structured templates. This includes details about training, license, limitations, etc.
- The paper should discuss whether and how consent was obtained from people whose asset is used.
- At submission time, remember to anonymize your assets (if applicable). You can either create an anonymized URL or include an anonymized zip file.

14. **Crowdsourcing and research with human subjects**

Question: For crowdsourcing experiments and research with human subjects, does the paper include the full text of instructions given to participants and screenshots, if applicable, as well as details about compensation (if any)?

Answer: [NA]

Justification: There were no human subjects involved.

Guidelines:

- The answer NA means that the paper does not involve crowdsourcing nor research with human subjects.
- Including this information in the supplemental material is fine, but if the main contribution of the paper involves human subjects, then as much detail as possible should be included in the main paper.
- According to the NeurIPS Code of Ethics, workers involved in data collection, curation, or other labor should be paid at least the minimum wage in the country of the data collector.

15. **Institutional review board (IRB) approvals or equivalent for research with human subjects**

Question: Does the paper describe potential risks incurred by study participants, whether such risks were disclosed to the subjects, and whether Institutional Review Board (IRB) approvals (or an equivalent approval/review based on the requirements of your country or institution) were obtained?

Answer: [NA]

Justification: There were no human subjects involved.

Guidelines:

- The answer NA means that the paper does not involve crowdsourcing nor research with human subjects.
- Depending on the country in which research is conducted, IRB approval (or equivalent) may be required for any human subjects research. If you obtained IRB approval, you should clearly state this in the paper.
- We recognize that the procedures for this may vary significantly between institutions and locations, and we expect authors to adhere to the NeurIPS Code of Ethics and the guidelines for their institution.
- For initial submissions, do not include any information that would break anonymity (if applicable), such as the institution conducting the review.

16. **Declaration of LLM usage**

    Question: Does the paper describe the usage of LLMs if it is an important, original, or non-standard component of the core methods in this research? Note that if the LLM is used only for writing, editing, or formatting purposes and does not impact the core methodology, scientific rigorousness, or originality of the research, declaration is not required.

    Answer: [NA]

    Justification: We only use LLMs for spell check and grammar correction.

    Guidelines:

    - The answer NA means that the core method development in this research does not involve LLMs as any important, original, or non-standard components.
    - Please refer to our LLM policy (`https://neurips.cc/Conferences/2025/LLM`) for what should or should not be described.

