# OpenReview forum: "Mellow: a small audio language model for reasoning"
_NeurIPS.cc/2025/Conference — NeurIPS 2025 poster_

### Official Review · Reviewer_Ae2v · 2025-06-07

**Clarity:** 3
**Significance:** 3
**Originality:** 3
**Rating:** 4
**Confidence:** 4

**Summary:**

This paper introduces Mellow, a small audio-language model (ALM) designed for reasoning tasks. Mellow achieves competitive or superior performance compared to many larger models, including some with over 50× the parameters and training data. The authors also release ReasonAQA, a new dataset combining synthetic and existing data to support audio-grounded reasoning. Extensive evaluations are conducted across multiple tasks—deductive reasoning, comparative reasoning, audio captioning, and multimodal understanding—along with thorough ablation studies.

**Questions:**

NA

**Ethical Concerns:**

["NO or VERY MINOR ethics concerns only"]

**Limitations:**

yes

**Quality:**

3

**Strengths And Weaknesses:**

Strengths:

- Timely and Relevant Contribution: Addresses an underexplored but important niche: reasoning with small ALMs, which is valuable for edge devices.

- Strong Empirical Results: Mellow outperforms all other small ALMs and even some large ALMs (e.g., Qwen-Audio) on reasoning tasks like MMAU and Audio Entailment.

- Novel Dataset (ReasonAQA): The construction of ReasonAQA demonstrates thoughtful design and use of LLMs for QA generation grounded in audio.

- Comprehensive Evaluation: Covers a wide spectrum of reasoning capabilities (deductive, comparative, binary QA, captioning), including OOD evaluation.

- Thorough Ablation Studies: Provides valuable insights on design choices (projection layers, LM architecture, audio encoder) and training strategies (fine-tuning vs. LoRA, etc.).

- Resource-Efficient: Only 167M parameters and trained on ~152 hours of audio, making it suitable for practical deployment.

Weakness:

- Speech Reasoning Gap: Mellow performs poorly on speech-based tasks due to lack of speech data in training. This limits generalizability.

- Synthetic Data Dominance: 70% of ReasonAQA is synthetic; concerns may arise around the diversity and generalizability of such data.

- Comparison with SOTA Models: While results are impressive, the evaluation would benefit from comparison with recent speech-capable models on more speech-focused tasks.

---

> ### Author Rebuttal · Authors · 2025-07-30
>
> We sincerely thank the reviewer for recognizing our contribution and novelty! We hope that our response resolves your concerns. Any follow-up questions are welcome.
>
> **Mellow performs poorly on speech-based tasks ... in training. This limits generalizability.** \
> We thank the reviewer for requesting clarification. Mellow is not trained on ASR data and therefore cannot perform speech recognition tasks. As a result, its ability to understand conversation-related information is limited. However, this is consistent with prior ALM literature, where models have focused primarily on audio and music understanding rather than speech content. For example, Pengi (NeurIPS 2023), LTU (ICLR 2024), and GAMA (EMNLP 2024) were all trained solely on audio and music data and thus exhibit similar limitations in speech understanding. While speech-based tasks are important, our current work is aligned with the audio-focused research direction and provides sufficient evidence for the claims made.
>
> **70% of ReasonAQA is synthetic; concerns may ... and generalizability of such data**\
> We appreciate this observation and emphasize that we took deliberate steps to ensure the quality and diversity of synthetic data. Specifically, we: (1) reduced the number of generated questions per caption to improve quality; (2) generated both detailed and MCQ formats to cover varied reasoning skills; (3) explicitly enforced audio grounding in prompts to avoid world-knowledge-only questions; and (4) instructed models to incorporate acoustic properties for richer audio-specific reasoning. Evidence of improved quality includes vocabulary analysis and grounded responses (Appendix Figures 6 and Table 13). Furthermore, 30% of ReasonAQA consists of curated reasoning datasets from the literature, ensuring diversity. The effectiveness of this dataset is shown by strong OOD performance on MMAU and MMAR benchmarks, therefore showing good generalization.
>
> **While results are impressive, the evaluation would benefit ... on more speech-focused tasks**\
> We acknowledge this point. Our primary focus is on non-speech audio reasoning, consistent with prior ALMs, and our evaluations therefore emphasize sound and music reasoning benchmarks. Nonetheless, following the reviewer's suggestion, we additionally fine-tuned QwenAudio -- a speech-capable model -- on ReasonAQA and report results below:
> | Experiment | MMAU performance |
> |---|---|
> | Pengi | 30.19 |
> | QwenAudio - Zero Shot | 41.86
> | QwenAudio - LoRA | 45.90 |
> | QwenAudio - full finetune| 43.72 |
>
> Even after fine-tuning, Mellow remains competitive and, in many cases, outperforms QwenAudio despite being over 50× smaller. Evaluating against speech-focused reasoning benchmarks is a promising direction for future work as we explore adding speech reasoning capabilities to Mellow.

---

> ### Author Response · Authors · 2025-08-05
> **Request to review the rebuttal**
>
> Thank you reviewer Ae2v for reviewing our paper. We have addressed your questions in our rebuttal response. As the rebuttal period is nearing its conclusion, we kindly request you to review our rebuttal and share any additional comments or concerns you may have. Thank you once again for your valuable feedback!

---

### Official Review · Reviewer_9EBg · 2025-07-03

**Clarity:** 3
**Significance:** 2
**Originality:** 2
**Rating:** 3
**Confidence:** 4

**Summary:**

The paper introduces Mellow, a small (167M parameter) Audio-Language Model (ALM) designed specifically for reasoning tasks. Unlike prior work that relies on scaling model and data size, Mellow focuses on improving reasoning abilities through architectural and training innovations. It is trained on a new dataset, ReasonAQA, which blends real and synthetically generated audio-grounded question-answer pairs. Despite its small size and limited data (~152 hours of audio), Mellow achieves state-of-the-art performance among small ALMs and outperforms several much larger models on multiple benchmarks, including multimodal reasoning (MMAU), deductive reasoning (Audio Entailment), comparative reasoning (Audio Difference), and audio captioning. The paper also conducts extensive ablation studies, revealing the importance of language model pretraining, fine-tuning strategies, audio encoder quality, and targeted synthetic data in boosting reasoning performance without relying on large-scale data.

**Questions:**

* The comparison in Table 5 may not be entirely fair. If I understand correctly, the linear probe is clearly not a strong baseline, and the modeling capacity of a language model is naturally expected to perform better.
* The paper emphasizes the high quality of ReasonAQA and compares it with OpenAQA by analyzing its shortcomings. However, it does not clearly explain how the data generation process for ReasonAQA ensures such high quality. Is controlling the prompts alone truly sufficient to produce significantly better data?

**Ethical Concerns:**

["NO or VERY MINOR ethics concerns only"]

**Final Justification:**

Through rebuttal, the authors further elaborated on the performance comparison of Mellow against other models on the MMAU benchmark and clarified the data collection procedure for training. However, I remain concerned about the novelty of the proposed method and its actual applicability in resource-constrained scenarios. I believe that if the paper had introduced some innovations in model architecture design or training strategies, it would have been a strong candidate for acceptance. Therefore, I decide to maintain my initial score.

**Limitations:**

Yes

**Quality:**

3

**Strengths And Weaknesses:**

* Strengths:
    * The structure of this paper is fairly clear, and the experiments are thorough. Despite the relatively small model size, it still achieves good performance.
    * The paper conducts ablation studies from multiple perspectives, providing valuable insights for researchers who are new to the field.

* Weaknesses:
    * My main concern is that the paper lacks sufficient novelty. The methods used have already become relatively common practices in the field. Furthermore, the ablation studies do not introduce particularly novel analytical perspectives, and many of the conclusions drawn are largely aligned with what is already widely recognized by researchers. This paper feels more like a detailed experimental report rather than a research paper.
    * The paper attempts to justify the development of SALM from the perspective of applications for edge devices. However, it lacks analysis specific to resource-constrained scenarios. As a result, it fails to demonstrate the advantages of the proposed method in such contexts—for example, through comparisons with alternative solutions such as distilling large models into smaller ones or directly quantizing large models.
    * The fairness of the experimental comparisons in the paper is questionable. Although Mellow shows strong performance on the benchmarks tested, it seems more like an indication that Mellow may be overfitted to these specific tasks rather than a demonstration of general effectiveness. Perhaps showcasing the model’s performance on entirely unseen tasks would better demonstrate the generalization ability of the proposed approach.

---

> ### Author Rebuttal · Authors · 2025-07-30
>
> We thank the reviewer for recognizing our contributions and providing constructive feedback. We address every question and hope that our response resolves your concerns. Any follow-up questions are welcome.
>
> **My main concern is that the paper lacks sufficient novelty ... a research paper.**\
> We thank the reviewer for the comment. To provide context from the literature, one of the first ALMs released was Pengi, published at NeurIPS 2023. Pengi was a small audio-language model that achieved state-of-the-art performance at the time. In subsequent years, research in the field primarily focused on increasing the parameter count and training data for ALMs, leading to performance improvements and a shift from small to large audio-language models. However, the necessity of large ALMs has rarely been questioned. Fundamental questions -- such as what are the limits of small audio-language models? How far can we push their performance? What architectural changes improve small ALMs? Do these changes scale as more data is added? -- remain largely unexplored. We believe that answering these questions is essential to understanding both small and large ALMs and to building efficient small models. Therefore, we start with Pengi, the only small ALM in the literature, and subsequently improve upon it while keeping the model size comparable.
>
> We only partially agree with the statement that “many of the conclusions drawn are largely aligned with what is already widely recognized by researchers.” Small Audio-Language Models (ALMs) have not been previously explored prior to this work. Although the mechanisms or architectures we use may be common, our findings are novel and not previously reported in the literature. For example, we found that larger, transformer-based projection layers -- commonly used in prior work -- only improve performance when the language model is frozen. In contrast, with full fine-tuning, simpler projection layers perform better. Our ablation studies reveal a clear performance trend for training the language model backbone: full fine-tuning > LoRA adaptation > frozen LM. This challenges the common practice of using large language models with LoRA adaptation (as done in LTU, Qwen, GAMA, etc.). Moreover, while transformer-based projection layers consistently outperform linear and non-linear projections when the language model is frozen (as in Pengi), this advantage completely disappears when the language model is trainable. In that case, simple linear or non-linear projections achieve comparable performance. This contradicts the widely held assumption that complex projection layers are universally beneficial. Overall, we believe this line of work -- focused on understanding and systematically studying small ALMs -- is as important as traditional notions of novelty that emphasize introducing new components. While some individual ablations may not be entirely unique, it is necessary to study them specifically in the context of small audio-language models before making definitive claims.
>
> **The paper attempts to justify the development of ... smaller ones or directly quantizing large models.**\
> We thank the reviewer for bringing up this valid point. We agree that A key motivation for SALM is enabling edge deployment and reducing inference costs. Scaling from 8B to 135M parameters improves tokens-per-second throughput, resulting in faster responses, support for lower-compute devices, and reduced cloud latency and costs. In the NLP domain, this approach has already proven viable with models such as Phi, SmolLM, and MobileLLM. For concrete numbers, we evaluated Mellow and existing ALMs on an Nvidia V100 GPU, where each model was tasked with captioning 100 randomly sampled audio files from the AudioCaps test set. Throughput was computed as the number of tokens produced per second. The results are summarized below:
>
> | Model            | Parameters  | Total Time (s) | Total Tokens | Throughput (tokens/second) |
> |------------------|-------------|----------------|--------------|--------------------|
> | Qwen-Audio Chat  | 8.4B        | 52.99          | 3,039        | 17.04              |
> | GAMA             | 7B          | 199.30         | 3,593        | 18.03              |
> | Audio-Flamingo   | 2.2B        | 93.70          | 1,254        | 13.38              |
> | Mellow           | 167M        | 148.40         | 5,224        | 35.21              |
>
> Higher throughput reflects better performance, with Mellow significantly faster than 7B models. Moreover, models like Qwen-Audio 2 cannot fit on a V100 GPU and require larger memory. We believe that pursuing this research direction will lead to performant yet efficient ALMs.
>
> **The fairness of the experimental comparisons ... ability of the proposed approach.**\
> We thank the reviewer for this question. We use the MMAU benchmark, which consists of 10k annotated Audio Question Answering (AQA) pairs spanning the speech, sound, and music domains. This benchmark evaluates a model’s capabilities across 27 distinct skills involving various reasoning tasks. Notably, Mellow is trained exclusively on audio files from AudioCaps and Clotho, whereas MMAU sources its audio from 10 different datasets, with no overlap with Mellow’s training data. This makes MMAU an ideal benchmark for evaluating Mellow’s reasoning performance in an out-of-distribution (OOD) setting. However, we agree with the reviewer that it is still possible to overfit to MMAU.
>
> To address this concern, we additionally benchmark Mellow on the MMAR benchmark. This benchmark consists of 1,000 examples spanning sound, music, speech, and their mixtures, and it focuses on multi-step, deep reasoning beyond surface-level understanding. Moreover, MMAR was released after Mellow, making it a completely independent benchmark with no possibility of overfitting. The results are shown below:
> | Models| Size| Sound (%) | Music (%) | Speech (%) | Sound-Music (%) | Sound-Speech (%) | Music-Speech (%) | Sound-Music-Speech (%) | Avg (%) |
> |---|---|---|---|---|----|---|---|---|---|
> | **Audio Flamingo** | 2.2B   | 32.73 | 21.84 | 24.83 | 18.18 | 30.28  | 24.39 | 25.00 | 26.60|
> | **Audio Flamingo 2**| 1.0B| 20.61| 20.39| 24.15| 27.27| 23.85| 26.83| 25.00| 23.00|
> | **Audio Flamingo 2**| 2.5B| 26.67| 20.87| 22.79| 9.09| 22.94| 23.17| 20.83| 22.90|
> | **Audio Flamingo 2** | 4.7B  | 24.85 | 17.48 | 20.75 | 18.18 | 26.61| 23.17| 8.33 | 21.90|
> | **LTU** | 7B| 19.39| 19.90| 13.95| 18.18| 24.77| 21.95| 16.67 | 19.20|
> | **LTU-AS** | 7B| 20.00| 14.08| 19.05| 9.09| 20.64| 28.05| 12.50 | 19.00|
> | **MusiLingo** | 7B| 9.09| 7.28| 4.08| 9.09| 6.88| 7.32| 8.33 | 6.6 |
> | **MU-LLaMA**| 7B| 13.94| 13.59| 14.97| 9.09| 12.39| 14.63| 16.67 | 13.90|
> | **GAMA**| 7B| 29.09| 24.27| 27.89| 27.27| 24.77| 28.05| 20.83 | 26.50|
> | **GAMA-IT** | 7B| 22.42| 16.02| 12.24| 36.36| 22.48| 14.63| 12.50  | 17.40|
> | **Qwen-Audio-Chat**| 8.4B| 27.88| 20.39| 22.11| 9.09| 25.23| 25.61| 20.83 | 23.50|
> | **Qwen2-Audio** | 8.4B| 33.94| 23.30| 32.99| 9.09| 33.03| 26.83 | 33.33 | 30.40|
> | **Qwen2-Audio-Instruct**| 8.4B| 33.33| 24.27| 32.31| 9.09| 31.19| 30.49| 25.00 | 30.00|
> | **SALMONN**| 7B| 30.91| 29.61| 34.35| 9.09| 37.61| 28.05 | 37.50 | 32.80|
> | **SALMONN**| 13B| 30.30| 31.07| 34.69| 9.09| 34.86| 35.37| 41.67 | 33.20|
> | **Mellow** | 167M| 33.33| 26.70| 24.83| 18.18| 37.16| 32.93| 29.17 | 30.00|
>
> Mellow performs comparably to state-of-the-art reasoning models in the literature while being only one-tenth their size. Specifically, it outperforms all models except SALMONN, remaining competitive with Qwen2 and demonstrating strong performance on sound- and music-based questions despite having only 167M parameters.
>
> **The comparison in Table 5 may not be entirely fair ... expected to perform better.**\
> We thank the reviewer for the question. We use linear-probe performance because it is the standard fine-tuning procedure used in the Audio Entailment paper (AAAI 25). We followed the same method to ensure comparability.
>
> **The paper emphasizes the high quality of ReasonAQA and ... sufficient to produce significantly better data?**\
> We thank the reviewer for the clarifying question. The full data generation process is described in Appendix Sections C and D; we provide a concise summary here. Below are the some steps we took to ensure high-quality synthetic data generation:
>
> (1) Reduced Question Count: We deliberately reduced the number of generated questions per caption from 10 (as in OpenAQA) to 5, significantly improving question quality. As noted in Section D.1, requiring 10 questions per caption in OpenAQA often led to illogical or poorly structured questions. (2) Varied Question Formats: We generated both detailed and multiple-choice (MCQ) questions, each serving different learning objectives. MCQs enforce precise answer selection and reasoning about incorrect options, while detailed questions encourage rich, descriptive responses. (3) Explicit Audio Grounding: We added specific instructions such as “you are listening to the audio instead of reading the captions” and “generate question-answer pairs about what you just heard” (Figure 5). This discourages world-knowledge-only questions that can be answered without audio (e.g., “Q: What kind of sound is sizzling? A: A hissing sound when frying food”), a common issue observed in OpenAQA (Section D.1, Table 12). (4) Acoustic Property Emphasis: We instructed the model to explicitly use words related to acoustic properties -- semantic relations, spectro-temporal characteristics, frequency, loudness, duration, materials, interactions, and sound sources. We also used few-shot prompting with audio-focused examples.
>
> Evidence of the improved synthetic data quality includes vocabulary analysis (Figure 6), showing richer, more audio-specific terminology, and comparative examples (Table 13), demonstrating more grounded and detailed responses.

---

> > ### Comment · Reviewer_9EBg · 2025-08-06
> > **follow-up**
> >
> > I appreciate the authors’ patient response; however, I still have reservations regarding the novelty of the manuscript.

---

> ### Author Response · Authors · 2025-08-05
> **Request to review the rebuttal**
>
> Thank you reviewer 9EBg for reviewing our paper. We have addressed your questions in our rebuttal response. As the rebuttal period is nearing its conclusion, we kindly request you to review our rebuttal and share any additional comments or concerns you may have. Thank you once again for your valuable feedback!

---

### Official Review · Reviewer_rXRV · 2025-07-05

**Clarity:** 3
**Significance:** 2
**Originality:** 2
**Rating:** 4
**Confidence:** 4

**Summary:**

This paper introduces Mellow, a small audio language model for audio reasoning. The paper also introduces ReasonAQA dataset for training Mellow. ReasonAQA is built based on AudioCaps and Clotho data. Experiments show that training on ReasonAQA enables Mellow to achieve relatively good performance on the MMAU benchmark.

**Questions:**

- Did the authors try to use ReasonAQA data for fine-tuning audio LLMs such as SALMONN or QwenAudio? Will this approach yield more remarkable results?
- How does Mellow perform on other audio benchmarks besides MMAU?

**Ethical Concerns:**

["NO or VERY MINOR ethics concerns only"]

**Final Justification:**

The authors' rebuttal provides results on the MMAR benchmark to show the robustness of Mellow. Therefore, I will raise my score to 4.

**Limitations:**

See Weaknesses.

**Quality:**

2

**Strengths And Weaknesses:**

Strengths:
- The Mellow model with only 167M parameters can achieve promising performance on the MMAU benchmark.

Weaknesses:
- About the novelty. The model structure seems to merely replace a large language model with a small one, which is somehow trivial, and the model's improvement seems to stem from the ReasonAQA data. It is not surprising to obtain better results through training with higher-quality data. Therefore, I believe the novelty of this paper is limited.
- Does Mellow understand speech? I believe it cannot understand speech content at all, as the annotations in AudioCaps and Clotho don't include any conversation-related information. If that's the case, the lack of speech understanding ability in Mellow also limits its novelty.
- Regarding the rationality of the results. I am concerned that many baseline models are underestimated. Most models in Table 4 have scores below 60, but Mellow exceeds 91, which is rather questionable. Could you provide the results of some state-of-the-art (SOTA) models for reference, such as Qwen-Omni or Gemini? In addition, as for the caption performance of Table 4.4, the caption results of some models are lower than what I expected. For example, the SPICE score of SALMONN-13B is less than 10, which is weird.  When testing audio LLMs, did the authors adjust different prompts?  Some audio LLMs are quite sensitive to prompts, and inappropriate prompts could lead to lower results. The authors should adjust the test prompts to avoid underestimating the performance of the tested models.

---

> ### Author Rebuttal · Authors · 2025-07-30
>
> We sincerely thank the reviewer for the detailed comments. We reiterate the key contributions of our work: this paper presents Mellow, a small ALM capable of reasoning over audio and text, outperforming much larger models on several benchmarks (MMAU, Entailment, Difference Explanation, Captioning, AQA) while using only ~152 hours of training audio. We also introduce ReasonAQA, a 1M-instance dataset with curated reasoning questions grounded in audio content. Finally, our ablation studies reveal the effectiveness of architectural choices and training strategies in enabling reasoning in compact ALMs, and we present several additional findings.
>
> We have carefully addressed each of the reviewer’s concerns below. Any follow-up questions are welcome.
>
> **About the novelty. The model structure seems to ... the novelty of this paper is limited.**\
> We thank the reviewer for the comment. To provide context from the literature, one of the first ALMs released was Pengi, published at NeurIPS 2023. It was a small ALM that achieved state-of-the-art performance at the time. In subsequent years, the field largely shifted toward increasing parameter count and training data for ALMs, leading to performance improvements and a move from small to large ALMs. However, during this progression, the necessity of large models was rarely questioned. Fundamental questions -- such as what are the limits of small audio-language models? How far can we push their performance? What architectural changes improve small ALMs, and do these changes scale with additional data? -- remain largely unexplored. We believe answering these questions is crucial for understanding both small and large ALMs and for building efficient models. Therefore, we start with Pengi, the only small ALM in the literature, and subsequently improve it while keeping the model size comparable.
>
> We disagree with the sentence *"the model’s improvement seems to stem from the ReasonAQA data ... through training with higher-quality data."* First, high-quality data is indeed important, and we believe ReasonAQA is a valid, novel contribution to the field, particularly since AQA datasets are currently very limited. Second, data is only one factor contributing to model performance. For example, Pengi -- the only small ALM in the literature -- achieves 3.40 on the MMAU benchmark. To test the effectiveness of ReasonAQA, we trained a Pengi-like model using ReasonAQA data alone. This model achieves a score of 30.19 on MMAU, as shown in Table 8 (line 3). We then made subsequent improvements to the projection layer, pretraining strategy, SLM fine-tuning, audio encoder, and data augmentation techniques, ultimately reaching an updated score of 48.40 on MMAU. This 18-point absolute improvement is solely due to the architectural and training improvements proposed in the paper.
>
> **Does Mellow understand speech? I believe ... speech understanding ability in Mellow also limits its novelty.**\
> We thank the reviewer for requesting clarification. Mellow is not trained on ASR data and therefore cannot perform speech recognition tasks. As a result, its ability to understand conversation-related information is limited. However, this is consistent with prior ALM literature, where models primarily focus on audio and music understanding rather than speech content. For example, Pengi (NeurIPS 23), LTU (ICLR 24), and GAMA (EMNLP 24) are all trained solely on audio and music data, resulting in similarly limited speech understanding capabilities. While conversation-related abilities are desirable, our current work aligns with the existing audio literature and provides sufficient evidence to support the claims being made.
>
> **Regarding the rationality of the results ... the test prompts to avoid underestimating the performance of the tested models.**\
> We understand the reviewer’s concern and thank them for raising this valid point. The numbers and benchmarks reported in Table 4 are borrowed from the Audio Entailment paper (Table 4), published at EMNLP 2024. These are standard literature-reported results with prompts optimized for each task by the respective paper authors. Since these represent state-of-the-art results, we report them directly without further prompt optimization.
>
> **Did the authors try to use ReasonAQA data for fine-tuning ... more remarkable results?**\
> We thank the reviewer for the question. We used ReasonAQA to retrain Pengi, which is the only model with a parameter count comparable to Mellow. The results are shown in Table 8, line 3. The retrained Pengi achieves an MMAU score of 30.19, whereas Mellow achieves a score of 48.40. We did not evaluate QwenAudio or SALMONN in the submitted paper. However, following the reviewer’s suggestion, we fine-tuned QwenAudio on the ReasonAQA dataset using both LoRA fine-tuning and full fine-tuning (with no frozen parameters). The results are shown in the table below:
> | Experiment | MMAU performance |
> |---|---|
> | Pengi | 30.19 |
> | QwenAudio - Zero Shot | 41.86 |
> | QwenAudio - LoRA | 45.90 |
> | QwenAudio - full finetune| 43.72 |
>
> The full fine-tuning version performs worse than the LoRA fine-tuned model. Upon examining model responses, we see that the fully fine-tuned QwenAudio heavily overfits the ReasonAQA dataset, likely due to the mismatch between the dataset size and QwenAudio’s parameter count. In contrast, LoRA fine-tuning allows the model to adapt to ReasonAQA while retaining its broader world knowledge. Scaling ReasonAQA to better match QwenAudio’s parameter count might improve performance, but this is beyond the scope of the current work.
>
> **How does Mellow perform on other audio benchmarks besides MMAU?**\
> We appreciate the reviewer's suggestion and assistance in improving our work. We benchmark Mellow on several other audio benchmarks, including Audio Entailment (deductive reasoning), Audio Difference (comparative reasoning), Audio Captioning, and Audio Question Answering. The results and detailed explanations are provided in Sections 4.2, 4.3, and 4.4, respectively. We specifically focus on MMAU benchmark, which consists of 10k annotated AQA pairs spanning the speech, sound, and music domains. This benchmark evaluates a model’s capabilities across 27 distinct skills involving various reasoning tasks. Notably, Mellow is trained exclusively on AudioCaps and Clotho, whereas MMAU sources its audio from 10 different datasets, with no overlap with Mellow’s training data. This makes MMAU an ideal benchmark for evaluating Mellow’s reasoning performance in an out-of-distribution setting.
>
> However, to address concerns about overfitting or in-distribution training, we also evaluate Mellow on the MMAR benchmark. It consists of 1k examples spanning sound, music, speech, and their mixtures, and focuses on multi-step, deep reasoning beyond surface-level understanding. Moreover, MMAR was released after Mellow, making it a completely independent benchmark with no possibility of overfitting. The results are shown below:
> | Models| Size| Sound (%) | Music (%) | Speech (%) | Sound-Music (%) | Sound-Speech (%) | Music-Speech (%) | Sound-Music-Speech (%) | Avg (%) |
> |---|---|---|---|---|----|---|---|---|---|
> | **Audio Flamingo** | 2.2B   | 32.73 | 21.84 | 24.83 | 18.18 | 30.28  | 24.39 | 25.00 | 26.60|
> | **Audio Flamingo 2**| 1.0B| 20.61| 20.39| 24.15| 27.27| 23.85| 26.83| 25.00| 23.00|
> | **Audio Flamingo 2**| 2.5B| 26.67| 20.87| 22.79| 9.09| 22.94| 23.17| 20.83| 22.90|
> | **Audio Flamingo 2** | 4.7B  | 24.85 | 17.48 | 20.75 | 18.18 | 26.61| 23.17| 8.33 | 21.90|
> | **LTU** | 7B| 19.39| 19.90| 13.95| 18.18| 24.77| 21.95| 16.67 | 19.20|
> | **LTU-AS** | 7B| 20.00| 14.08| 19.05| 9.09| 20.64| 28.05| 12.50 | 19.00|
> | **MusiLingo** | 7B| 9.09| 7.28| 4.08| 9.09| 6.88| 7.32| 8.33 | 6.6 |
> | **MU-LLaMA**| 7B| 13.94| 13.59| 14.97| 9.09| 12.39| 14.63| 16.67 | 13.90|
> | **GAMA**| 7B| 29.09| 24.27| 27.89| 27.27| 24.77| 28.05| 20.83 | 26.50|
> | **GAMA-IT** | 7B| 22.42| 16.02| 12.24| 36.36| 22.48| 14.63| 12.50  | 17.40|
> | **Qwen-Audio-Chat**| 8.4B| 27.88| 20.39| 22.11| 9.09| 25.23| 25.61| 20.83 | 23.50|
> | **Qwen2-Audio** | 8.4B| 33.94| 23.30| 32.99| 9.09| 33.03| 26.83 | 33.33 | 30.40|
> | **Qwen2-Audio-Instruct**| 8.4B| 33.33| 24.27| 32.31| 9.09| 31.19| 30.49| 25.00 | 30.00|
> | **SALMONN**| 7B| 30.91| 29.61| 34.35| 9.09| 37.61| 28.05 | 37.50 | 32.80|
> | **SALMONN**| 13B| 30.30| 31.07| 34.69| 9.09| 34.86| 35.37| 41.67 | 33.20|
> | **Mellow** | 167M| 33.33| 26.70| 24.83| 18.18| 37.16| 32.93| 29.17 | 30.00|
>
> Mellow performs comparably to state-of-the-art reasoning models in the literature while being only one-tenth their size. Specifically, it outperforms all models except SALMONN, remaining competitive with Qwen2 and showing strong performance on sound- and music questions.

---

> > ### Comment · Reviewer_rXRV · 2025-08-06
> >
> > The authors' rebuttal has addressed my concerns, and I will raise my score to 4.

---

> ### Author Response · Authors · 2025-08-05
> **Request to review the rebuttal**
>
> Thank you reviewer rXRV for reviewing our paper. We have addressed your questions in our rebuttal response. As the rebuttal period is nearing its conclusion, we kindly request you to review our rebuttal and share any additional comments or concerns you may have. Thank you once again for your valuable feedback!

---

### Official Review · Reviewer_n6cA · 2025-07-05

**Clarity:** 3
**Significance:** 3
**Originality:** 3
**Rating:** 5
**Confidence:** 3

**Summary:**

This paper advances the understanding of building efficient ALMs with regard to trade-offs between model size, data constraints, and performance. While authors do not employ novel, groundbreaking techniques, I think this represents solid technical work with a good and important direction for the LLM-reasoning community. Although given the speech performance and limited audio knowledge, the claims regarding immediate applicability require further consideration.

**Questions:**

Please see the weakness section above. All questions are embedded with the weakness I found.

**Ethical Concerns:**

["NO or VERY MINOR ethics concerns only"]

**Limitations:**

yes.

**Paper Formatting Concerns:**

none.

**Quality:**

4

**Strengths And Weaknesses:**

Strengths:
Amidst the trend towards scaling LLMs to billions of parameters, I think this paper is well-motivated towards closing the gap by showing that their 135M model is able to achieve competitive reasoning over its bulkier counterparts.

Overall the paper is well-written and clear to understand. The evaluation methods are well-explained and cover multiple reasoning types {deductive, comparative, ood}

Important ablations are provided across tuning methods (lora, finetuning, prefix), data scaling effects with wav2caps, and prompting styles.

Weakenss:
While the authors have intentionally constrained to using only Audiocaps+clotho to isolate reasoning improvements from those of scaling parameters, I think this severely limits the model's knowledge of audio concepts and of course generalization.
Would a model with double Mellow's parameters but trained on the same constrained data still underperform due to insufficient audio concept coverage?
Would the simple projection and HTSAT encoder scale effectively to larger parameter counts? Is this reasoning ability transferable or clotho-specific pattern matching?
I think demonstrating reasoning improvements in isolation, may not provide sufficient evidence for the broad claim that small models can maintain competitive performance in practical applications where both larger models AND larger datasets are available

Could you please provide Fig. 3 to compute the drop against only music and sound subsets of MMAU? I think this should help validate “(1) Speech-based task constitute nearly 30% of MMAU … limited speech training ..” In my opinion, this is not just a limitation but a fundamental weakness for an audio understanding model.

While 135M parameters are indeed smaller than cloud-scale models, it still requires significant resource (>500 mb;  authors, please confirm this.). Most successful applications for on-device or even always-on are in 1-5M param range - considered truly on-device AI which occasionally trigger server-side computation for main LLMs. Given their param count, currently both Mellow and Qwen2 Audio sit in the same bucket - i.e both are for server side deployment.
If a device triggers a late-stage process, it already transmits the audio to the server. Given that the server hypothetically has no compute constraints, why would one opt for Mellow there instead of passing the audio to bulkier QewnAudio for potentially better results?
I think some clearer articulation of the intended deployment scenarios would immensely help this paper.

---

> ### Author Rebuttal · Authors · 2025-07-30
>
> We thank the reviewer for recognizing our technical contributions, thorough experiments, clear writing, and providing constructive feedback. We address every question and hope that our response resolves your concerns. Any follow-up questions are welcome.
>
> ---
> **While the authors have intentionally constrained ... performance in practical applications where both larger models AND larger datasets are available**\
> The reviewer raises a valid point regarding the choice of experimental setup and its implications. To reiterate the premise: in audio models, reasoning can improve through two approaches. First, expanding data (audio) coverage (e.g., knowledge of audio objects and scenes), which provides a broader foundation upon which the model can apply its existing reasoning capabilities. Second, enhancing the model’s inherent reasoning ability through architectural, learning, or post-training techniques. By fixing the training audio, one can conduct controlled evaluations of an ALM’s reasoning performance, which can later be scaled via data expansion. Therefore, we break down the reviewer’s question into two main parts: (1) do reasoning improvements achieved in isolation translate when trained on larger datasets? and (2) do models with smaller parameter counts retain competitive performance when larger datasets are available?
>
> For point (1), we agree with the reviewer that by constraining data, we limit the model's knowledge of audio concepts and its generalization ability. However, we believe that by identifying a minimal recipe on constrained data, the approach can later be scaled (in both data and parameters) to achieve better performance. One empirical piece of evidence supporting this is shown in Table 8 (the "+ WavCaps" row) and further discussed in Appendix F7. In this experiment, we utilize the popular WavCaps dataset and, using the same generation method as ReasonAQA, generate additional training data. This data provides broader coverage of audio concepts, and the results show an absolute 1.5\% improvement in overall MMAU performance. Notably, music-related performance improves by an absolute 10\%.
>
> For point (2), the WavCaps experiment provides some indication that improvements hold when the training data is increased. However, how competitive this approach would be compared to larger models trained on larger datasets remains an open question, and this work does not provide evidence to conclusively answer it. Unfortunately, due to resource constraints, we cannot scale the parameter count to conduct the necessary experiments for validation. Nevertheless, we believe this work is an essential step toward building efficient ALMs, which will eventually help close the performance gap between large models trained on massive datasets and smaller ALMs.
>
> **Could you please provide Fig. 3 ... weakness for an audio understanding model.**\
> We thank the reviewer for this suggestion. We conducted this experiment for Mellow, where we measured the drop in performance on the music- and sound-only subsets of MMAU. However, due to space restrictions, we did not include it in the paper. The results are shown in the table below:
> | Experiment | Audio Input | Gaussian Noise | Difference |
> |---|---|---|---|
> | Sound | 61.26 | 39.24 | 22.02 |
> | Music | 54.19 | 39.79 | 14.4 |
> | Speech | 29.73 | 28.70 | 1.03 |
> | Average | 48.40 | 35.91 | 12.49 |
>
> The results show that performance on sound and music drops significantly, while speech performance changes only marginally. This implies that the model actively utilizes audio information when answering sound and music questions but relies mostly on the language model for speech predictions. This behavior is expected since the model was not trained on speech data. Therefore, in the paper, we state: “(1) Speech-based task constitute nearly 30% of MMAU … limited speech training ..”, which explains why the overall average difference is not higher. We will add this discussion to the Appendix.
>
> **While 135M parameters are indeed smaller than cloud-scale models, ... of the intended deployment scenarios would immensely help this paper.**\
> We thank the reviewer for raising this point and helping to improve the manuscript. Mellow, while having 167M parameters, still occupies approximately 670 MB of storage. We agree with the reviewer that most successful on-device or always-on applications typically operate in the 1-5M parameter range. However, there are multiple instances where LLaMA-like models are successfully deployed on devices such as iPhones, laptops, and tablets after applying quantization and other optimization techniques.
>
> By scaling down from 8B to 135M parameters, we can significantly improve the tokens-per-second throughput of these models. This improvement translates to faster responses and better support for older devices on the edge side, while on the cloud side, it results in lower inference costs and reduced latency. Even in server-side deployments, where compute resources can theoretically scale infinitely, minimizing inference costs remains desirable. In the NLP domain, this approach has already proven viable with models such as Phi, SmolLM, and MobileLLM. Similarly, choosing Mellow or other small ALMs offers benefits for both edge and cloud scenarios. For concrete numbers, we evaluated Mellow and existing ALMs on an Nvidia V100 GPU, where each model was tasked with captioning 100 randomly sampled audio files from the AudioCaps test set. Throughput was computed as the number of tokens produced per second. The results are summarized below:
>
> | Model            | Parameters  | Total Time (s) | Total Tokens | Throughput (tokens/second) |
> |------------------|-------------|----------------|--------------|--------------------|
> | Qwen-Audio Chat  | 8.4B        | 52.99          | 3,039        | 17.04              |
> | GAMA             | 7B          | 199.30         | 3,593        | 18.03              |
> | Audio-Flamingo   | 2.2B        | 93.70          | 1,254        | 13.38              |
> | Mellow           | 167M        | 148.40         | 5,224        | 35.21              |
>
> A higher throughput indicates better performance. As shown, Mellow is significantly faster than the representative 7B model. Moreover, models like Qwen-Audio 2 cannot fit on a V100 GPU and require larger memory. We believe that pursuing this research direction will lead to performant yet efficient ALMs.

---

> ### Author Response · Authors · 2025-08-05
> **Request to review the rebuttal**
>
> Thank you reviewer n6cA for reviewing our paper. We have addressed your questions in our rebuttal response. As the rebuttal period is nearing its conclusion, we kindly request you to review our rebuttal and share any additional comments or concerns you may have. Thank you once again for your valuable feedback!

---

### Decision · Program_Chairs · 2025-09-17

**Decision:**

Accept (poster)

**Comment:**

This work reverses the trend in audio-LMs, by proposing a "small" (~170M params, compared to 8B in sota Qwen2) model which is performant comparable to sota models. This is achieved using:
(i) a new dataset comprising "reasoning" related question/answers obtained from LLMs on audio-text pairs from audio captioning datasets.
(ii) a number of training techniques, e.g., full fine-tuning instead of lora, transformer-based projection layers, better lm, representation learning using ssl.

All the reviewers appreciated the clear presentation, performant small ALMs, and thorough ablations.

Concerns were raised regarding: (i) lacking speech abilities, (ii) being task or data specific.
However, the authors fine-tuned sota models (e.g., Qwen) on the proposed dataset and showed they still underperform the proposed method, and, also showed good performance on new tasks (not included in training), and new datasets (e.g., MMAR released after the proposed method).

Given the above, this work may be valuable to the community to build on small models, especially as the authors promise to share their models and code. Hence, I am inclined to recommend this work for acceptance.